# Carbon storage through China's planted forest expansion

Kai Cheng [1,2,9], Haitao Yang [1,9], Shengli Tao [2], Yanjun Su [3,4], Hongcan Guan [5], Yu Ren [1,2], Tianyu Hu[3,4], Wenkai Li[6], Guangcai Xu[7], Mengxi Chen[1], Xiancheng Lu [1], Zekun Yang[1], Yanhong Tang[2], Keping Ma [3,4], Jingyun Fang[2,8] & Qinghua Guo [1,2] ✉

China's extensive planted forests play a crucial role in carbon storage, vital for climate change mitigation. However, the complex spatiotemporal dynamics of China's planted forest area and its carbon storage remain uncaptured. Here we reveal such changes in China's planted forests from 1990 to 2020 using satellite and field data. Results show a doubling of planted forest area, a trend that intensified post-2000. These changes lead to China's planted forest carbon storage increasing from $675.6 \pm 12.5$ Tg C in 1990 to $1,873.1 \pm 16.2$ Tg C in 2020, with an average rate of ~ 40 Tg C yr$^{-1}$. The area expansion of planted forests contributed ~ 53% ($637.2 \pm 5.4$ Tg C) of the total above increased carbon storage in planted forests compared with planted forest growth. This proactive policy-driven expansion of planted forests has catalyzed a swift increase in carbon storage, aligning with China's Carbon Neutrality Target for 2060.

Forests function as important carbon (C) sinks[1–6] and offer a natural solution to address climate change and ecological issues[7–9]. The expansion of planted forest areas is considered an effective solution to achieve the main objectives of the UN's Global Forest Goals under continuous decrease in natural forest area[10]. However, the decrease in natural forest area outweighs the increase in planted forest area, resulting in a global net decline of forest area by 178 million ha and a reduction in the global forest C stock from 668 Petagram (Pg) to 662 Pg between 1990 and 2020[11]. Achieving the goals of the 2030 Agenda[12] relies on credible monitoring and verification of the C storage of planted forests. This task needs to take into account land spatiotemporal dynamics and forest types, distinguishing between planted and natural forests due to different species composition, stand structure, age, and management context[13–16]. In China, which has the world's largest planted forest area, accounting for over 1/4 of global planted forest area[17], a comprehensive analysis of its planted forests C storage needs investigation.

China's achievement in expanding planted forest area is a remarkable accomplishment both domestically and in the global context[18]. Despite this global reduction of forest area, China's planted forest area expansion has made significant contributions to increasing China's forest area for several consecutive years[19,20]. As of 2020, China's forest area reached approximately 220 million ha, accounting for 5% of the world's forest area[11]. This expansion is a result of conducted government efforts to convert croplands, shrublands, and grasslands into planted forests[17,18,21,22]. This effort to grow the planted forest area not only increased China's total forest extent but also resulted in drastic changes in land use and land cover (LULC), impacting China's forests C storage capacity[5,11,23–26]. Based on the National Forest Management Plan (2016–2050), China's forest area will continue to account for ~26% of China's total land area by 2050, requiring an additional conversion of over 2.8 million ha from other LULC to forests[20]. This has highlighted the urgent need for accurately quantifying the spatiotemporal dynamics of planted forest conversion from

[1]Institute of Remote Sensing and Geographic Information System, School of Earth and Space Sciences, Peking University, Beijing 100871, China. [2]Institute of Ecology, College of Urban and Environmental Sciences, Peking University, Beijing 100871, China. [3]State Key Laboratory of Vegetation and Environmental Change, Institute of Botany, Chinese Academy of Sciences, Beijing 100093, China. [4]University of Chinese Academy of Sciences, Beijing 100049, China. [5]School of Tropical Agriculture and Forestry, Hainan University, Haikou 571737, China. [6]School of Geography and Planning, Sun Yat-Sen University, Guangzhou 510275, China. [7]Beijing GreenValleyTechnology Co. Ltd, Beijing 100091, China. [8]Key Laboratory for Earth Surface Processes of the Ministry of Education, Peking University, Beijing 100871, China. [9]These authors contributed equally: Kai Cheng, Haitao Yang. ✉e-mail: guo.qinghua@pku.edu.cn

other LULC types[13,27] and devising an optimal C storage strategy to take advantage of the planted forest expansion[28,29].

However, currently available maps of China's planted forests have limitations in both spatial and temporal coverage[30,31]. These existing national maps and datasets were created through forest inventories or the digitization of forest inventory maps for specific years with coarse spatial resolutions[21,32] or only pertain to specific subtypes[23,33,34], and are thus inadequate for tracking China's multi-decadal efforts in the planted forest area expansion[13,35,36]. Consequently, there is an urgent need to conduct high spatial resolution, national-scale research, and long-time-series assessments of C storage associated with the planted forest area expansion in China.

For this purpose, leveraging all available Landsat-4/5/7/8/9 surface reflectance images from 1990 to 2020 in the Google Earth Engine (GEE) cloud computing platform together with field samples, we generated the high-resolution wall-to-wall planted forest maps for China. These efforts yielded the ability to capture the dynamics of planted forest coverage at five-year intervals from 1990 to 2020 at 30-m spatial resolution (see Methods and Supplementary Information). Integrating the data on China's planted forests' spatiotemporal dynamics, LULC conversions, vegetation C densities, and vegetation type maps, we then analyzed China's planted forest C storage from 1990 to 2020. The resulting comprehensive dataset not only contributes to the understanding of the impact of China's planted forest

expansion but also serves as a valuable tool for evaluating its implications on climate change and sustainable development within the country.

## Results

### Mapping planted forest spatiotemporal dynamics in China

Quinquennial maps of China's planted forest from 1990 to 2020 reveal that the area of planted forests expanded at an annual rate of 14,613 km², representing a 94.33% net increase from 464,715 km² in 1990 to 903,099 km² by 2020 (Fig. 1a, b, e). This increase in planted forest area was the combined result of a gain of 479,681 km² and a loss of 41,297 km², leading to a net increase of 438,384 km² (Fig. 1e and Supplementary Fig. 1). Our maps were compared with planted forest map of National Forest Inventory and the reported National Forestry Statistical Yearbook, showing minor differences in the area and significantly positive relationships (R² ranged from 0.8 to 0.9, P < 0.01) (Supplementary Figs. 2 and 3). The overall accuracies for the generated quinquennial maps ranged from 77.3% to 81.8% from 1990 to 2020 (Supplementary Fig. 4). These accuracy assessments support the reliability of the generated maps of China's planted forests, providing confidence that our maps align with existing datasets and fulfill the aims of this study.

China's planted forests adhere to typical forest distribution patterns in terms of latitudinal, longitudinal, and elevational

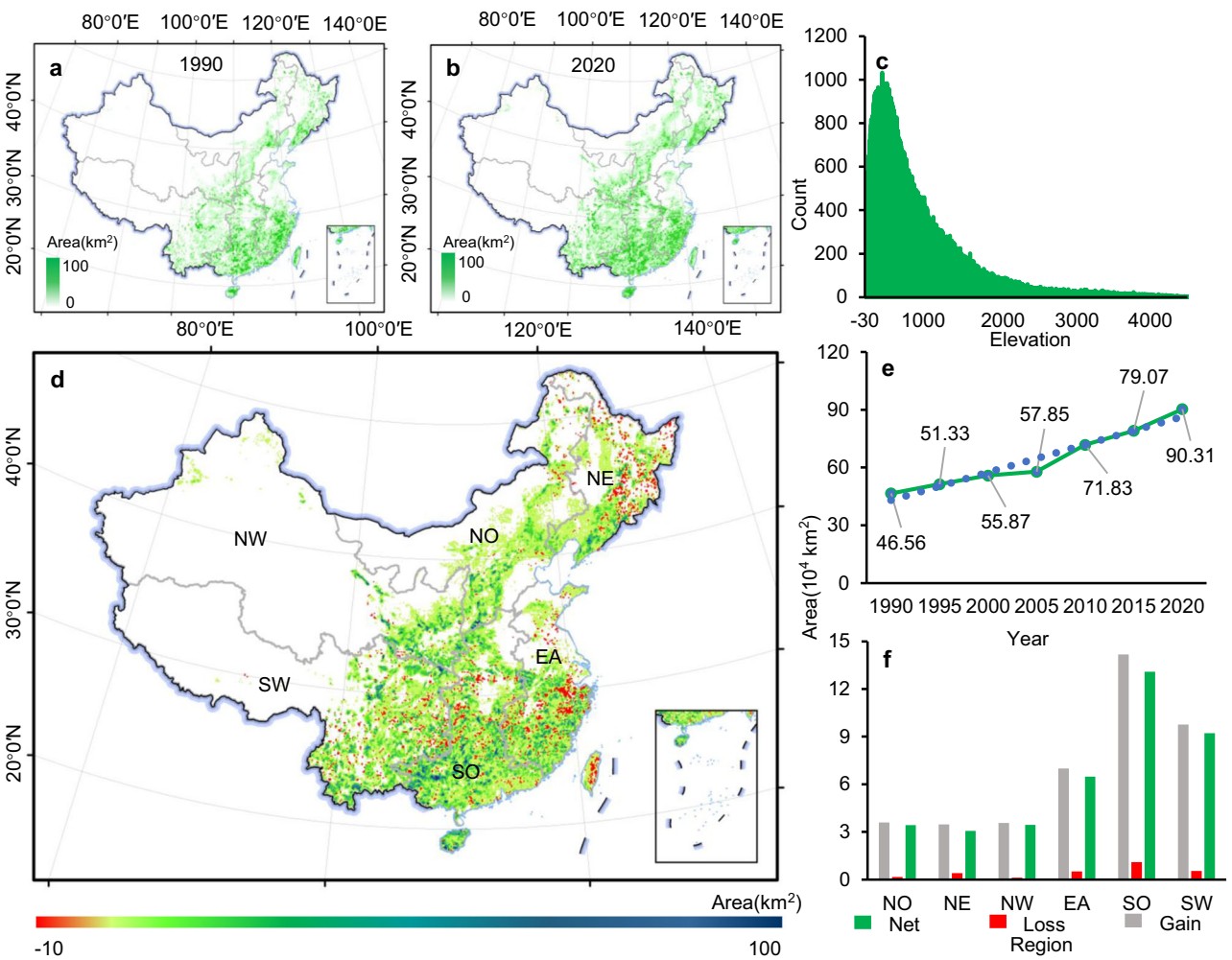

**Fig. 1 | Spatiotemporal dynamics of planted forest area from 1990 to 2020.**
**a**, **b** China's planted forest distribution in 1990 and 2020 in a 0.1° grid scale, respectively. **c** Distribution of the planted forests along with various elevations. **d** Spatial distribution of planted forest area gains and losses from 1990 to 2020 in a

0.1° grid scale. **e** Change in planted forest area every five years from 1990 to 2020. **f** Planted forest area gains and losses at six geographical regions from 1990 to 2020. NO north, NE northeast, NW northwest, EA east, SO south, SW southwest.

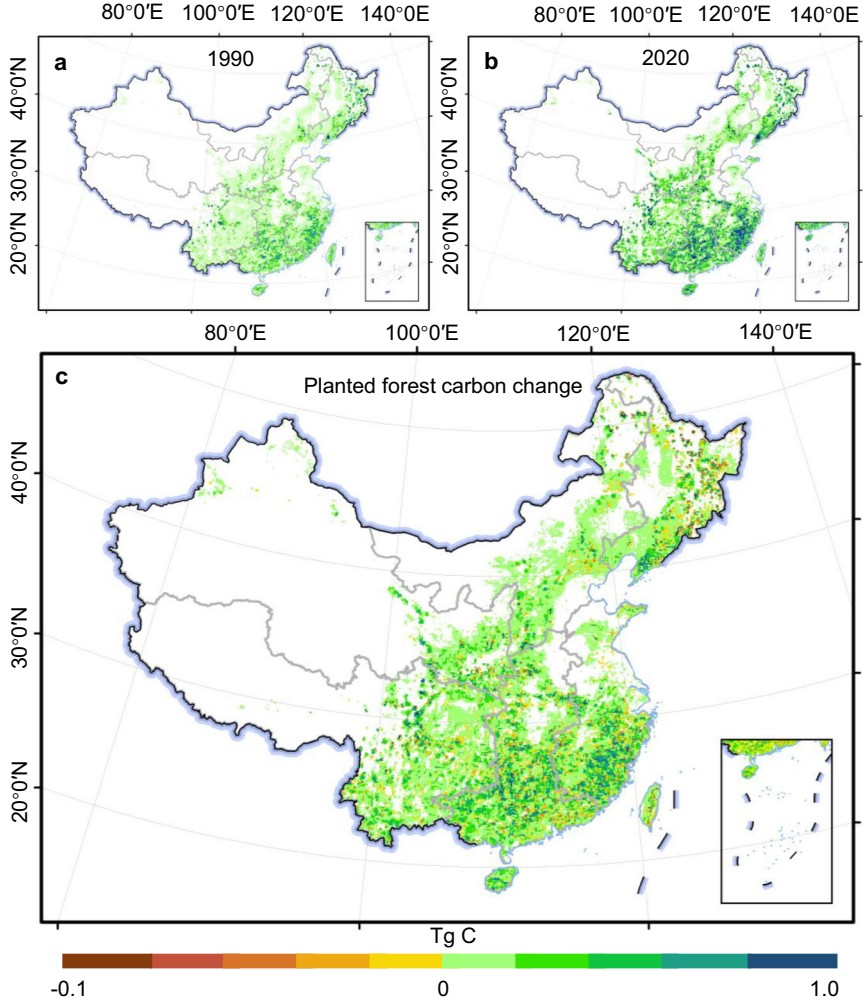

**Fig. 2 | Spatiotemporal changes of planted forest C storage in China. a** Map of China's planted forest C storage in biomass in 1990 in a 0.1° grid scale. **b** Map of China's planted forest C storage in biomass in 2020 in a 0.1° grid scale. **c** Map of China's planted forest C storage changes in a 0.1° grid scale (The changes in planted forest C storage within the grid in 2020 compared to the planted forest C storage in the same grid in 1990).

distribution[32]. In 2020, planted forests were predominantly concentrated in regions with elevations below 1,500 m (Fig. 1c). The majority of planted forests were located in the southern region, accounting for 32.6% of the total area of planted forests, followed by the eastern region (20.1%) and the southwestern region (16.6%) (Fig. 1a, b). The northwestern region, on the other hand, had the smallest proportion of planted forest area, covering only 5.5% of its total area (Fig. 1 and Supplementary Fig. 5), but had the fastest rate of planted forest area expansion, growing by 266.5% from 1990 to 2020 (Fig. 1f). Overall, all regions showed a net increase, albeit with significant fluctuations around 2005 (Fig. 1f and Supplementary Fig. 6b). The south experienced the highest increase rate, with a total increase of 150,088 km² and an average increase rate of 5,003 km² per year (Fig. 1f). The southwest (103,729 km²) and the east (73,208 km²) also exhibited substantial expansion (Fig. 1f). By aggregating the 30-meter resolution planted forest map into 0.1° grid cells, we analyzed the characteristics of planted forest area changes at the grid scale. We found that 91% of the changed pixels from 1990 to 2020 showed positive trends, with 84.0% of them being statistically significant ($P < 0.01$) (Supplementary Fig. 7a). These positive trends were mostly observed in the east and south (Supplementary Fig. 7a). In terms of the changing trends in the expansion rate of planted forest areas, we observed that only 7.4% of the changed pixels exhibited positive trends, with 24.4% being statistically significant ($P < 0.01$)

(Supplementary Fig. 7b), which were mainly distributed in the southwestern, northwestern, and northern regions (Supplementary Fig. 7b). In most regions of China, the rate of planted forest area expansion showed a decline, which is consistent with the trend observed in China's annual afforestation and reforestation area from 1990 to 2020, as reported in the National Forestry Statistical Yearbook (Supplementary Fig. 8), particularly in the eastern, southern, and northern regions[29], where the suitable area for plantation is nearing saturation (Supplementary Fig. 8).

## Changes in planted forest C storage
The dynamics of China's planted forest C storage in biomass from 1990 to 2020 exhibited marked spatiotemporal variations (Fig. 2). Overall, we found a trend of increasing C storage in China's planted forest, increasing from 675.6 ± 12.5 (mean ± standard deviation) Teragram (Tg) C in 1990 to 1,873.1 ± 16.2 Tg C in 2020, with an annual increment of approximately 40.0 Tg C (Fig. 3a and Supplementary Table 1). The increase in C storage of China's planted forests closely aligned with the spatiotemporal dynamics of planted forest distribution and expansion (Figs. 1d and 2c). Since 1990, the most substantial increases in C storage within China's planted forests have been observed in regions with extensive forest cover (Figs. 1 and 2). The increased C storage was primarily attributed to the ongoing expansion of the planted forest area. However, the growth of young planted forests also significantly

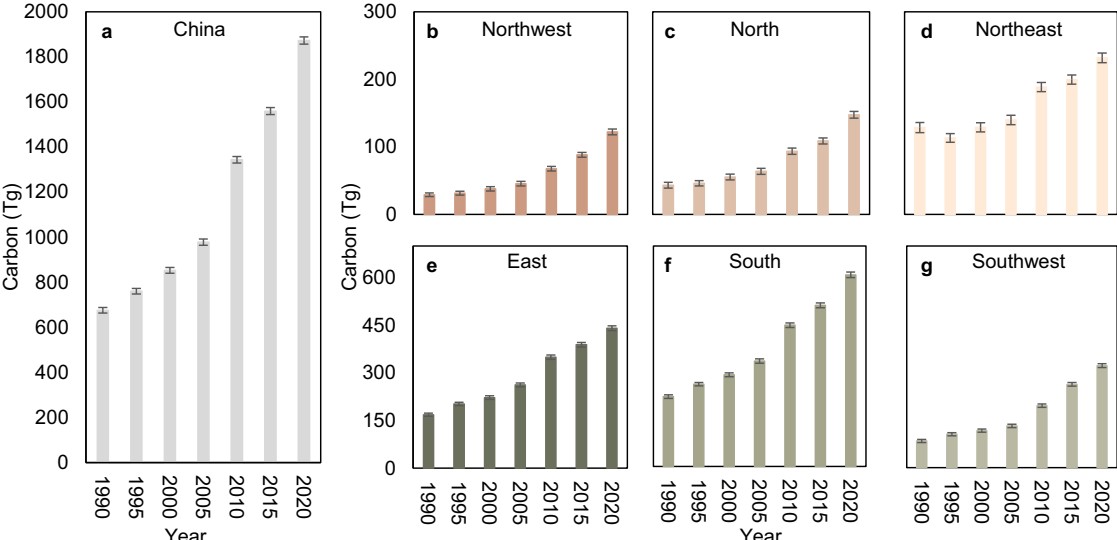

**Fig. 3 | Change in C storage in biomass of planted forests across various regions. a** C storage change of planted forests at a national scale from 1990 to 2020. **b–g** C storage changes of planted forests in the Northwest, North, Northeast, East, South, and Southwest of China (see Fig. 1 for the location of each region), respectively. The error bars are one standard deviation of the estimated mean C storage.

contributed to the increase of C storage (Fig. 3). Despite observed regional declines (Fig. 2c), the overarching trend of increasing C storage remains unabated (Figs. 2 and 3).

The C storage in planted forests in southern, eastern, and southwestern China increased the most, with respective gains of 386.4 ± 20.6 Tg C, 272.6 ± 18.1 Tg C, and 237.9 ± 15.8 Tg C from 1990 to 2020 (Fig. 3e–g). Despite an initial decline in C storage in the northeastern region between 1990 to 1995, there was an overall increase of 103.1 ± 20.4 Tg C by 2020 (Figs. 2c and 3d). The historically forest-scarce northern and northwestern (Fig. 1) regions exhibited increasing trends in C storage of planted forests, attributable to the expansion of planted forests, with increases from 43.3 ± 4.2 Tg C to147.8 ± 5.0 Tg C and from 29.2 ± 2.6 Tg C to 122.2 ± 4.2 Tg C in 1990 and 2020, respectively. Overall, the rate of increase in C storage in planted forests before 2005 was comparatively modest compared to the period following 2005 (Fig. 3a), averaging approximately 20 Tg C/a between 1990 and 2005 (Fig. 3a and Supplementary Table 1). However, the C storage rates in planted forests across different regions experienced a marked acceleration after 2005 (Fig. 3b–g), with an average storage rate of almost 60 Tg C/a from 2005 to 2020 and peaking at a sequestration rate of approximately 73 Tg C/a between 2005 and 2010 (Fig. 3a and Supplementary Table 1).

**Attributions of LULC transformation on planted forest C storage**
The spatiotemporal analysis revealed a synchronization between the expansion of planted forests from LULC conversion and the associated increase in planted forest C storage (Fig. 4). An analysis of LULC transformation between 1990 and 2020 indicated that 438,787 km² of planted forests originated from the conversion of croplands, shrublands, grasslands, and natural forests, representing 98.3% of the overall increase in planted forest area (Fig. 4a). During the same period, the conversion of different LULC to planted forests resulted in 637.2 ± 5.4 Tg C, approximately 53.2% of the overall increase of planted forest C storage (Supplementary Table 2). Nonetheless, the contribution from LULC conversion to the C storage of planted forests varied across different periods, with the highest C storage contribution occurring from 1990 to 1995, reaching 81.4%, and declining to mere a 17.2% from 2000 to 2005 (Supplementary Table 2). This variability can be attributed to the expansion rate of China's planted forests in distinct periods, along with the spatiotemporal variations in the types of LULC

converted into planted forests (Fig. 4b, c). Until 2000, expansions were mainly from natural forests, shrublands, and croplands (Fig.4a, c). Post-2000, conversions from cropland, shrublands, and grasslands emerged as the principal contributors (Fig. 4a, c). The classification of conversion events indicated that over one-third of single-change events involved transitions from croplands to planted forests, succeeded by conversions from shrublands, grasslands, and natural forests (see Methods and Supplementary Figs. 9 and 10). Multi-change events, accounting for 7.2% of all transitions, predominantly occurred in the southern and southwestern regions (Supplementary Figs. 9 and 10). The prevalent cropland-to-planted forest conversions in both single-change and multi-change events (Supplementary Figs. 9 and 10) highlight the impact of the Grain for Green (GFG) program initiated in 1999. Consequently, between 1990 and 2020, C storage gains from cropland-to-planted forest conversions amounted to 191.7 ± 2.6 Tg C, with shrubland and grassland conversions contributing 176.4 ± 2.3 Tg C and 135.9 ± 2.0 Tg C to planted forest C storage, respectively (Supplementary Table 2). C storage gains from the conversion of natural forests to planted forests were relatively lower, with an estimated 121.7 ± 3.5 Tg C, constituting about 19.1% of the total increase in C storage from LULC conversion (Supplementary Table 2).

We conducted a more detailed analysis of C storage changes due to LULC conversion into planted forests across various regions in China, revealing substantial regional disparities in the contributions of these conversions to augmenting C storage in planted forests (Fig. 5). In both the northwestern and northern regions, grassland conversion emerged as the predominant factor driving the increase in C storage within planted forests (Fig. 5). In the northwestern region, 34.5% of the C storage increase in planted forests from 1990 to 2020 was contributed by the conversion of grassland, while in the northern region, the contribution of grassland was 25.9% (Fig. 5 and Supplementary Table 3). In the northeastern region, conversions from cropland and natural forests to planted forests significantly contributed to the region's planted forest C storage increase, accounting for 27.4% and 19.1% of the total increase, respectively (Fig. 5 and Supplementary Table 3). The eastern region witnessed a notable increase in C storage in its planted forests, with cropland and natural forest conversions contributing 11.9% and 13.0% to the overall increase, respectively (Fig. 5 and Supplementary Table 3). In southern and southwestern China, the

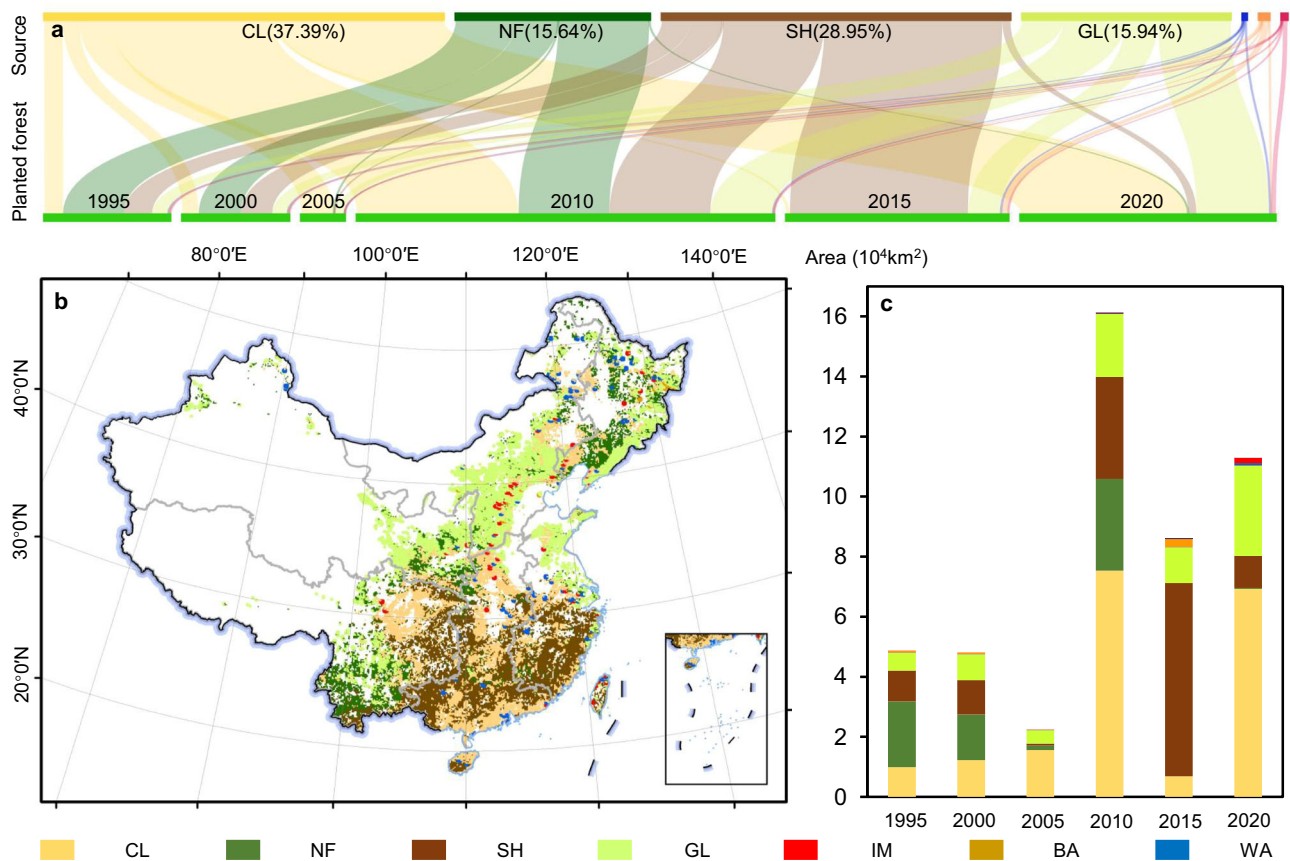

**Fig. 4 | LULC classes converted to planted forests in China from 1990 to 2020.**
**a** Flow pattern of conversion of main LULC types into planted forests across different periods. **b** Spatial distribution of LULC types converted to planted forests from 1990 to 2020. **c** Area and proportion of LULC types converted to planted forests in different periods. CL cropland, SH shrubland, GL grassland, NF natural forests, BA barren, WA water, IM impervious, SI snow and ice.

main conversions were from cropland and shrubland, especially in the southwestern region, where the conversion of shrubland contributed 25.3% to the increase in C storage of planted forests in this region, and might be the main reason for the sharp increase in C storage of planted forests from 2010 to 2015 (Fig. 5 and Supplementary Table 3).

## Discussion

Our high-resolution, quinquennial maps of China's planted forests unveiled the spatiotemporal dynamics of China's planted forests from 1990 to 2020, enabling an assessment of their ecological benefits, particularly the role in climate change mitigation through enhancement of terrestrial C storage through planted forests.

The expansion of China's planted forests has been widely attributed to China's policies[18]. China's economic boom has placed a heavy burden on the environment, with negative consequences on impoverished rural communities[37] and natural disasters such as floods and droughts[38]. These challenges have prompted the implementation of extensive afforestation/reforestation programs[18]. While there are regional differences in planted forest projects, their shared goal is to protect fragile environments and forests against water runoff, soil erosion, landslides, flooding, and desertification[17,19,20]. For example, from 1999, the implementation of the GFG program has led to an increase in forest and grass types on cropped hillslopes (>15°) and the conversion of croplands, barren hills, and wasteland to forests[19]. Until 2019, 40.5% of the planted forest area was a result of the GFG program[19]. Other programs, including the Three-North Shelterbelt Development Program (SDP-TN) and the Shelterbelt Development Program in Five Regions (SDP-FR), also have had significant effects on expansion of China's planted forest in different spatial and temporal scales[18,30].

However, we also need to acknowledge that large-scale mapping based on remote sensing, while powerful, may introduce some uncertainties due to the existing technology's limitation in recognizing smaller patches[39]. Additionally, the time it takes for planted forests to develop detectable canopy cover from seeds (~5-7 years) or seedlings (~3-5 years) is a factor to consider[40]. Therefore, the spatiotemporal dynamics of planted forests estimated based on our study may have a certain time lag, which could explain the rapid expansion of planted forests after 2005, as the policy-driven expansion of planted forests around 2000 may have been just detected and analyzed during this period. Overall, according to the National Forest Management Plan (2016–2050), the overall increasing trends of China's planted forest area are expected to persist[20], but the rate of increase may decline due to limitations in available land area for planted forests[29]. Our maps of China's planted forests timely capture the spatiotemporal dynamics of China's planted forests over the past three decades and can facilitate future forest protection and expansion.

Studies have demonstrated the reliability of forest C storage estimations when based on remote sensing techniques[41]. By integrating a combined approach of remote sensing and field surveys, we have not only accounted for the C storage brought about by the expansion of planted forest area from LULC conversion but also focused on the contribution of planted forest growth to C storage (see Methods). This further enhances the accuracy of China's planted forests C storage estimates[13–16]. The potential estimation errors in C storage arise from the C density and vegetation map used in this study. For example, the classification of planted forest types over various periods with the same vegetation map may present a challenge to our study because vegetation types have the potential to change. However, our previous

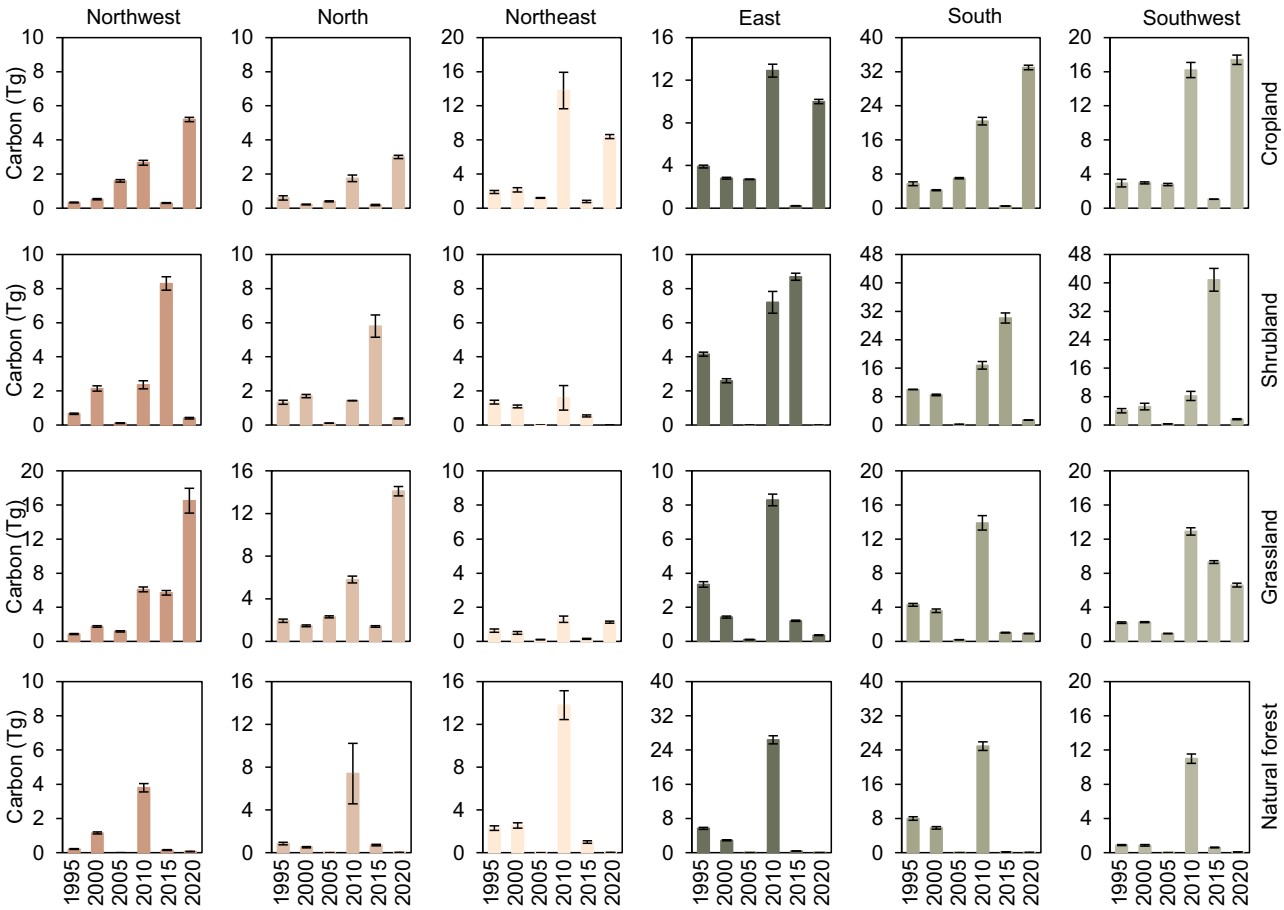

**Fig. 5 | Changes in C storage caused by the conversion of cropland, shrubland, grassland, and natural forest to planted forests at the regional scale.** The error bars are one standard deviation of the estimated mean C storage.

studies have indicated that vegetation types in China are stable[23,24], justifying our use of the same vegetation map to classify planted forest types across different periods. Recognizing that C density changes with forest age, we constructed a C density database based on the national forest inventory data, considering age-dependent C density variations (see Methods). Furthermore, the results of this study are consistent with those of previous research (e. g. Fang et al.[42]) on China's planted forest C storage, supporting the accuracy of our estimates.

From 1990 to 2020, the increase in China's planted forests C storage due to area expansion exceeded 50% of the total C storage increment in China's planted forests (Figs. 2 and 3). However, the growth contribution of young and newly established planted forests to C storage requires more attention[5,13]. The increase in C storage brought about by the growth of planted forest was also significant, especially in the eastern, southern, and northern regions, where the contribution of planted forest growth to C storage amounts to 60.4%, 46.8%, and 47.6% of the total C storage in these regions, respectively (Fig. 3). Results in the northeastern and southern regions were not surprising, due to the widespread distribution of planted forests from 1990 to 2020 (Fig. 1). The increase in C storage in the northern region may be attributed to growth of planted forests, which afforested by the SDP-TN[30]. In contrast, other regions mainly relied on LULC to planted forest conversions to gain C storage, particularly in the southwestern region, where a large amount of LULC conversion contributed to a 63.9% increase in C storage (Figs. 4 and 5).

Over the period from 1990 to 2020, planted forests in the eastern, southern, and southwestern regions of China accounted for about 75% (896.9 Tg C) of the total C storage by planted forests (Fig. 3). A contributing factor is the favorable climatic conditions in these regions, including ample water availability, which have accelerated both expansion and growth of planted forests[43,44]. Previous research indicated that in regions with high evaporation and transpiration rates (water-stressed regions), the survival rate and growth status of planted forests may be inadequate[45]. This explains why the distribution of planted forests in the northwestern and northern regions, despite their large geographical area, is not as extensive (Fig. 1). Another factor is the implementation of ecological restoration projects has significantly alleviated ecological issues in these areas, leading to a positive cycle and further promoting the expansion of planted forests in those regions. Notably, despite the conversion of over 4.1 million ha of croplands to planted forests[19], domestic crop production in China increased by 44.0% between 2000 and 2018[46]. A third possible contributing factor is the potential capacity of young planted forests to promote rapid increases in C storage[5,6,13]. The net C storage of forest shows a distinctive trend, with a rapid increase in young age, a peak in middle age, and a decline in old age[47]. This characteristic of forests explains why China's planted forests have been experiencing a period of a rapid C storage increment over the past three decades, which has become more pronounced since 2005 (Figs. 3 and 5).

Compared to the eastern, southern, and southwestern regions of China, the increase in C storage of the planted forests in the northwestern and northern regions has been relatively slow, with increase rates of only 3.1 Tg C/a and 3.5 Tg C/a, respectively (Fig. 3). Despite innovative afforestation techniques such as runoff forestry and deep planting, which integrate seedling and container-based methods, are actively being pursued to enhance the drought resistance and survival rates of planted forests[48], the potential C storage capacity of planted forests in northern and northwestern regions is still lower than in

eastern, southern, and southwestern regions[26,31]. Concern about China's planted forests in the northern and northwestern regions have persisted for many years, as the rapid and extensive expansion of planted forests in certain regions may come at the expense of other ecological functions[39] and could lead to water shortages[49]. Given these concerns, further research is warranted to comprehensively address the negative impacts brought about by the expansion of planted forests in these areas. As a result, we recommend that management strategies should evolve from mere afforestation to a holistic approach that encompasses protection, management, and utilization, with an emphasis on the importance of planted forest tending.

Our study characterized the spatiotemporal dynamics of C storage in the biomass of China's planted forests over a thirty-year period, from 1990 to 2020. We found that the increase in C storage within China's planted forests can be primarily attributed to the expansion of planted forest areas. This expansion has largely occurred through the conversion of croplands, shrublands, and grasslands into planted forest areas. In addition to area expansion, the growth of China's planted forests themselves represents another significant factor contributing to C storage. Post-2000, a policy-driven push for expanding planted forest areas has led to rapid increase, with the majority of these planted forests currently in their young or middle age. Based on these results, we anticipate that the newly expanded planted forests will possess substantial C storage potential in the future, potentially playing a key role in helping China meet its Carbon Neutrality Target by 2060.

## Methods
### Definitions of natural and planted forests in this study
In this study, forests are defined as areas with tree height exceeding 5 meters and canopy cover exceeding 15% within a 30-meter resolution pixel, planted forests refer to a forest ecosystem established by artificial tree planting or seeding for the provision of income and goods, as well as for climate change mitigation and the restoration of ecosystem services and processes[17,50]. Natural forests refer to a forest that regenerates naturally, with trees growing from naturally dispersed seeds or seedlings, and is typically composed of indigenous tree species and genotypes[11,51]. In this study, we only focused on the spatiotemporal dynamic of China's planted forest and resultant C storage in aboveground biomass from 1990 to 2020.

### Landsat data
We utilized all available Landsat TM/ETM/OLI/OLI-2 surface reflectance images for China from January 1, 1988, to December 31, 2021, leveraging the GEE cloud-processing platform. A total of 447,730 atmospherically corrected Landsat scenes, with a cloud cover of less than 30%, were selected from Landsat-4 (1,186 scenes), Landsat-5 (160,830 scenes), Landsat-7 (192,645 scenes), Landsat-8 (91,468 scenes), and Landsat-9 (1,601 scenes). These scenes were used to generate wall-to-wall planted forest maps at five-year intervals from 1990 to 2020 (Supplementary Table 4). At each time step, we created a stack of images for the target year ± two years (e.g., images from 1988 to 1992 were used for compositing the 1990 mosaic) to minimize missing data and mitigate cloud interference[52]. Finally, we applied the C function of mask (CFMASK) algorithm, implemented in the GEE platform, to every stack to mask out clouds, cloud shadows, and snow[53].

### Field samples
The annotated samples of planted forests at various periods formed the basis for generating time-series maps of planted forests in this research. Initially, this research acquired more than 600,000 vegetation survey samples through crowdsourcing, field surveys, and national forest inventories. These samples contain information such as geographic coordinates and vegetation types[36], covering the period from 2003 to 2022. With the aid of these samples, our research employed a time-series change detection method to establish a dataset for the recognition of planted forests at different time intervals by evaluating if each sample experienced changes during different time periods. First, based on the vegetation type information of each sample, we selected samples of planted and natural forests. Second, we employed the spatial analysis tool in ArcGIS Pro 3.0 to analyze the forest mask data from the annual China Land Cover Dataset (CLCD)[54] from 1990 to 2020 (Supplementary Note 1), resulting in the identification of forest regions that remained unchanged during this period. Third, using the identified unchanged forest regions, we applied spatial filtering to all planted and natural forest samples, preserving only the samples that intersected with the unchanged forest areas, thereby obtaining an initial set of field samples suitable for different time intervals. Fourth, we used the Normalized Difference Vegetation Index (NDVI) time series from Landsat to determine if the minimum NDVI value for each sample in the created dataset from 1990 to 2020 was equal to or greater than 0.6[55], retaining samples that satisfied this condition. By following these procedures, we acquired a dataset containing 124,407 samples of planted and natural forests, with 70% allocated for model training and 30% for validation (Supplementary Fig. 11).

### Mapping planted forests and validation
Significant differences exist in textural and temporal features between natural and planted forests. Previous research has demonstrated that planted forest mapping can be achieved by utilizing these differences in conjunction with a large number of samples at the national scale[36]. Therefore, we first derived five widely used vegetation indices from Landsat-based surface reflectance data, including the NDVI, enhanced vegetation index (EVI), bare soil index (BSI), soil-adjusted vegetation index (SAVI), and modified soil-adjusted vegetation index (MSAVI). These indices were appended to the cloud-free image stacks, along with the spectral bands (blue, green, red, near-infrared, shortwave infrared-1, and shortwave infrared-2), to facilitate planted forest extraction (Supplementary Note 2). Second, we computed the median value of each pixel to composite the final image (11 bands). According to the final composite image, textural information was calculated through the gray-level co-occurrence matrix method in GEE, which generated 18 features for each band. The temporal features were calculated by analyzing Landsat NDVI and EVI time series from 1990 to 2020 through harmonic analysis[56]. Specifically, the magnitude, phase, amplitude, and root mean square error (RMSE) of the fitted NDVI and EVI time series were extracted to constitute a temporal feature set[56,57] (Supplementary Note 2). Additionally, topographic information (elevation, slope, and aspect) was retrieved through digital elevation model (DEM) data from the Shuttle Radar Topography Mission (Supplementary Note 2). In summary, 220 features were used for mapping planted forests, including 11 spectral bands, 198 textural features, 8 temporal features, and 3 topographic features (Supplementary Data 1). To reduce the number of features used for planted forest mapping, we applied the recursive feature elimination cross-validation (RFE-CV) feature selection method at the regional scale (Supplementary Note 3) and established the optimal feature collection for the northeastern, northern, northwestern, southern, and southwestern regions at different periods (Supplementary Fig. 12). Third, a classification model adapted to each region (Supplementary Data 1) and period was constructed according to a random forest classifier and used to identify planted forests, respectively. Finally, we generated planted forest maps for the years 1990, 1995, 2000, 2005, 2010, 2015, and 2020 (Supplementary Fig. 1).

The resultant time-series planted forest maps were validated through four data sources (Supplementary Note 4). First, we used 30% of the independent field samples to evaluate the maps (Supplementary Note 4 and Supplementary Fig. 11). Second, a systematic random

sampling was implemented to verify the generated planted forest maps (Supplementary Note 4 and Supplementary Fig. 13). We also collected statistical data on planted forest area at national and provincial scales for the period 2000-2020 and compared resultant maps with the statistical data (Supplementary Note 4). Furthermore, the digitized planted forest map based on the Seventh National Forest Resource Inventory conducted from 2004 to 2008[21] was further obtained and used to compare the mapped results for the years 2000, 2005, and 2010 (Supplementary Note 4). The validation results indicated that the generated planted forest maps had a comparable accuracy and exhibited a high degree of concordance with existing maps and statistical data at national and provincial scales (Supplementary Figs. 2, 3, and 4).

## Planted forest spatiotemporal dynamic analysis

To gain a comprehensive understanding of the spatiotemporal dynamics of planted forests in China, this research initially aggregated the planted forest areas for each period into a 0.1° grid and subsequently analyzed the changes of planted forest area, the area added every five years, and their trends at national, regional and grid scales using simple linear regression with a $t$-test at the 5% significance level based on the "lm" function in the R software.

Furthermore, we employed the image change detection tool in ArcGIS Pro 3.0, based on the CLCD land cover data corresponding to 1990, 1995, 2000, 2005, 2010, 2015, and 2020, to analyze land cover changes resulting from the expansion of planted forest area. The focus was to identify the land cover types (e.g., cropland, grassland, etc.) that underwent conversion to planted forests in each period.

Lastly, we conducted a statistical analysis of the change frequency for each pixel within the changing areas, dividing them into single-change events and multiple-change events. Single-change events refer to occurrences in which a pixel experienced only one change during the six periods of 1990-1995, 1995-2000, 2000-2005, 2005-2010, 2010-2015, and 2015-2020, while multiple-change events involve pixels that underwent two or more changes. This analysis aids in identifying hotspots of planted forest change in China during the last 30 years.

## Planted forest C storage

This study used China's 1:1,000,000 vegetation type map to estimate the C storage of planted forests in biomass using the C density method. To accurately estimate C storage in planted forests over different periods, we collected the seventh national forest inventory data (representing 2005 in this study), which records tree species, geographical location, and age information. Based on the forest inventory data, we determined C densities for various types of planted forests at different periods. We then estimated changes in C storage in aboveground biomass of planted forests by comparing C storage in patch locations with the same vegetation type (see Fig. 2), following the process outlined below:

(1) Acquisition of planted forest inventory samples for different forest types. We extracted planted forest samples from the forest inventory data based on forest type annotations. The planted forest samples were divided into 17 subsets (Supplementary Table 5) according to the forest types reported in China's vegetation map.

(2) Acquisition of planted forest samples across various periods. We used the age information from the forest inventory data and remote sensing time-series analysis to construct planted forest inventory samples for the various vegetation types corresponding to different periods. Since the forest inventory data used in this study represent 2005, for 1990 to 2000, the samples were directly acquired according to the age variation in the inventory data. For instance, the inventory samples with a tree age exceeding 15 years represent they did not have a disturbance between 1990 and 2005, and can therefore be used to represent the 1990 sample, whose age is obtained by subtracting 15 years. For 2010 to 2020, we first used NDVI and forest time-

series masks to assess the changes in each sample from 2005 to 2010, 2005 to 2015, and 2005 to 2020, preserving those that remained unchanged and discarding any that had altered. By employing these methodologies, we acquired inventory samples for each period, inclusive of information about the tree species and age.

(3) Estimation of C density. With the inventory sample collections established for each period, the age-biomass density equation (Eq. 1) at the tree species scale crafted by Xu, et al.[58] was applied to compute the biomass density for each sample in all forest types, and the C density value for each sample was determined using the 0.5 biomass-carbon conversion factor suggested for China's forests by Ma, et al.[59]. Subsequently, the average C density and its standard deviation for each type were calculated. It is important to note that this approach might overstate the C density of newly established planted forests between 2005-2020, as their ages ranged from 1-15 years, and the sample ages for 2010, 2015, and 2020 determined were all above 5 years. Consequently, for the newly added planted forests in this period, the study adopted the widely used space-for-time substitution approach[60], selecting 2005 samples aged 1-15 years to estimate the C density of these new types of planted forests using the age-biomass equation (Eq. 1):

$$B = \frac{w}{1 + ke^{-at}} \quad (1)$$

in which $B$ is the biomass density (Mg C/ha), $t$ is the forest age, $w$, $k$, and $a$ are constants that were determined by Xu, et al.[58].

(4) Calculation of C storage in planted forests. According to the forest types of the vegetation map, we first classified the planted forest area into 17 forest types based on the nearest neighbor principle. Utilizing the C densities of these vegetation types corresponding to disparate periods established (Supplementary Table 5), the C storage of China's planted forests for these periods was calculated as (Eq. 2):

$$C = \sum_{i=1}^{N} D_i \times Area_i \quad (2)$$

where $C$ is the total C storage in the biomass of the planted forests whereas $D_i$ and $Area_i$ are the C density and area of the $i^{st}$ planted forest type. $N$ is the number of types.

(5) Validation. To demonstrate the reasonableness of the calculated C storage in the planted forests, we used publicly accessible field survey samples created by Xu, et al.[61] (download from the National Ecological Data Center resource sharing service platform of http://www.nesdc.org.cn/) to validate the estimated C storage in 2010, because these samples were collected around in 2010. Additionally, we downloaded China's forest biomass data for the year 2019 generated by Yang, et al.[62] (https://www.3decology.org/2023/08/02/china-forest-agb-map2019/) and the Global dataset of forest aboveground biomass for the year 2020 (https://climate.esa.int/en/projects/biomass/data/) to validate the estimated C storage in 2020. The comparison results indicated an $R^2$ ranging from 0.50 to 0.68(Supplementary Fig. 14).

## Sources of errors and uncertainties

This research, together with our previous works[23,24,36,62,63], revealed the dynamics of China's planted forests and their C storage. However, some uncertainties remain in the results, mainly associated with mapped planted forest area and C density derived for vegetation types. Identifying them can help clarify the caveats associated with this study.

Errors in planted forest mapping originate from various sources. First, while the Landsat time series covers our study period adequately, the presence of cloud cover and shadows, combined with the extensive study area, made it difficult to obtain high-quality, cloud-free images for the entire country. This is the primary reason why our study could not generate annual planted forest maps. Therefore, to

maximize mapping accuracy, we created cloud-free and shadow-free composites every five years. Nevertheless, even after cloud and shadow removal, some gaps still exist, with the largest being only 0.2% (Supplementary Fig. 15), which is almost certain not to affect the classification results. Second, our classification process constructed the optimal feature set for each period out of the 220 features we established using the feature selection method (Supplementary Note 3). Although our classification features encompassed spectral, textural, and terrain information, we may have overlooked variables that could impact the final classification outcome. Furthermore, we did not conduct a comparative analysis of different classification algorithms and chose to use the widely accepted RF algorithm. However, there might be other machine learning algorithms better suited for planted forest identification. Third, our classification samples originated from three sources: national forest inventory, field sample, and crowdsourcing. National forest inventory data are widely recognized as the most authoritative source[1,58]. Field samples were collected by forestry experts from various provinces, and the accuracy of their labels is generally reliable and commonly used for remote sensing mapping[62]. However, the crowdsourced data collection method presented challenges for our classification process. Although each crowdsourced sample underwent expert verification[63], the subjectivity involved in determining pixel reference classes through photo interpretation by different experts may have introduced noise into the training and validation datasets[52]. However, this noise is expected to have a minimal impact on the final classification results[64]. Furthermore, we established samples for each period based on the NDVI time series of each sample, which may be affected by NDVI noise. To quantify the uncertainties of our resultant maps, we mapped the distribution of pixel uncertainty (Supplementary Note 5 and Supplementary Fig. 16), revealing that low and medium uncertainties exist in large areas of the dataset (>80%) (Supplementary Fig. 16). Less than 20% of the total area in the southwestern and northeastern regions was classified with relatively high uncertainties (Supplementary Fig. 16).

Errors in C storage estimation primarily stem from the following aspects. First, we used a 1:1,000,000 scale vegetation map of China to classify planted forests into 17 types for each period. However, the vegetation map used in this study was created between the 1980s and the 2000s, and over several decades, China's vegetation formations have undergone significant changes[24]. Therefore, there might be some misclassification when using this map for vegetation type classification. Nevertheless, the 1:1,000,000 scale China vegetation map represents the potential distribution regions of Chinese vegetation[65]. Although some species may have changed over the years, vegetation types and their general distribution have remained relatively stable compared to the 2020 updated vegetation map[23]. Hence, in this study, we employed a nearest-neighbor approach to classify planted forests into 17 types according to the forest types of vegetation map and then calculated the C storage. We also compared the classification results with the updated 2020 China vegetation map[23], which revealed that over 80% of grids had the same vegetation types (Supplementary Fig. 17). For C storage estimates, only 2% had a difference greater than 0.1 (Supplementary Fig. 17). Second, we estimated C densities for different forest types using national forest inventory data and referring to the age-biomass density equations developed by Xu, et al.[58]. We generated vegetation type C density datasets for different periods by considering forest age and change detection methods based on the forest inventory data. Although the forest inventory data used in our study were collected between 2004 and 2008, for consistency with our research data, we treated them as representative of the year 2005. As a result, this may introduce an error of 1-3 years in forest age, impacting the C density estimation. Additionally, this method might have resulted in an overestimation of C density for newly established planted forests after 2005. This is because, after 2005, the minimum forest age

for each period (including 2010, 2015, and 2020) exceeded 5 years. Therefore, we adopted a space-for-time approach to select samples with forest ages between 1 and 15 years in 2005 and calculated the C density for these newly established planted forests after 2005.

Finally, to quantify the uncertainty of C estimation, we calculated the mean and standard deviation of the C density of each forest type based on the forest inventory samples. The mean value was used to estimate C storage, and then we calculated the total uncertainty by calculating the C storage in planted forests using the lower and upper standard deviation values to provide a measure of uncertainty associated with the total C storage of planted forests[66].

## Data availability

The planted forest area mapped in this study is available from https://www.3decology.org/2024/04/15/chinas-planted-forest-maps-from-1990-to-2020/. The forest masks used are extracted from the China land cover dataset (CLCD), which can be downloaded from https://zenodo.org/records/5210928. The vegetation map of China can be obtained from the Data Center for Resources and Environmental Sciences, Chinese Academy of Sciences: http://www.resdc.cn.

## Code availability

The code for the statistical analysis of planted forest area dynamics is accessible on https://github.com/ChenYL2021/GuoLab.git.

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

## Acknowledgements

This study was supported by the International Research Centre of Big Data for Sustainable Development Goals (No. CBAS2022GSP06 (Q.H.G.)), the National Key Research and Development Program (No. 2022YFF13002002 (Q.H.G.), 2022YFF1300203(K.C.)), and the National Natural Science Foundation of China (No. 42371329 (Q.H.G.), and 32301285 (H.T.Y.)).

## Author contributions

Q.H.G., K.C., and H.T.Y. conceived and designed the project. K.C. performed the planted forest mapping, validation, and analysis. K.C. and H.T.Y. performed carbon calculations and drew the figures. K.C. and H.T.Y. wrote the first draft together. S.L.T. and Y.J.S. extensively edited the manuscript and provided suggestions to revise the manuscript. H.C.G., Y.R., T.Y.H., W.K.L., and G.C.X. provided reviewed this manuscript and provided suggestions for the language. M.X.C., X.C.L., and Z.K.Y., conducted field sampling. Y.H.T., K.P.M., and J.Y.F. supervised this study and provided essential suggestions. All authors contributed to the revision of the manuscript.

## Competing interests

The authors declare no competing interests.
