## [Peer Review File · Nature Communications]

Carbon storage through China's planted forest expansionEditorial Note: Parts of this Peer Review File have been redacted as indicated to remove third-party material where no permission to publish could be obtained.

REVIEWER COMMENTS

Reviewer #1 (Remarks to the Author):

This manuscript addressed the quantification of areal extent and carbon storage of planted forests in China from 1990 to 2020. The detection of planted area is achieved with Landsat images. Classification and validation of the newly planted forest maps are supported by an extensive dataset of sample measurements and auxiliary datasets. Dynamics of land cover are analyzed and attributed to afforestation and reforestation programs. The planted forest maps are then related to measurements from the National Forest Inventory to quantify their carbon storage and their potential uptake of carbon.

The manuscript is in principle of interest to readers of Nature Communications because it provides a detailed explanation of land cover trajectories in China during the last 30 years. It is acknowledged that this is a massive study, supported by many figures and tables in the Supplement. It is a strenuous task to bring all the evidence together in a single manuscript. However, I find the manuscript overloaded with topics, with a tendency towards the remote sensing component, which in turn makes the manuscript less appealing to this journal in my opinion. Aspects related to the implications of the findings are less pronounced, giving the feeling that the manuscript has not reached its natural balance yet.

Based on my background on remote sensing, I also see several aspects that need to be clarified concerning the derivation of the forest planted maps, their accuracy and the quantification of the carbon storage. To appreciate whether the numbers provided are reasonable, a thorough analysis of errors is needed in addition.

The authors find below a set of general comments that may stimulate some thinking on how the manuscript shall be improved and a more detailed list of comments having the aim of polishing the text. The authors are strongly encourage to revise their manuscript and submit

a new version to Nature Communications.

General comments

The aim apparently was to monitor the carbon storage dynamics. This aspect is vaguely addressed as it is not clear how C was validated and what is the magnitude of the error bars (text on lines 349-357 is not satisfactory).

The mapping sequence is not entirely clear. This is my understanding:

1) A forest mask for each epoch was created from existing land cover maps. Question here is how was “forest” defined. Could it be that the definition behind each dataset was different, leading to different accuracy statistics which then influenced the selection of the reference for the forest mask?

2) Planted forest and natural forest maps (yearly? for every five years?) were obtained from Landsat metrics + auxiliary datasets in correspondence of the forest mask. When evaluating these maps, user/producer accuracies are provided for the two classes but the manuscript discusses conversions from shrubland, cropland and other classes to/from natural and planted forest. How were the non-forest classes mapped? What are the accuracy statistics for these classes?

A flowchart would very much help.

Several comparisons are mentioned throughout the manuscript but no figure compares values (see also detailed comment for line 122). The number of figures could actually be reduced if reference data and mapped data were co-plotted. This would avoid jumping back and forth in the manuscript and let the reader appreciate how well the mapped data corresponds to the reference.

The Section on “Present and future planted C forest storage potential related to water availability” has certain merits but seems to be out of scope in this manuscript and should be reconsidered. I would focus on a transparent presentation of the planted forest maps (strengths and weaknesses), and reinforce the text describing the C balance of planted forests with implications in terms of policy and future developments with respect to future

forest expansion programs (if any of course).

The title of the Section “The largest LULC conversion in human history” is not acceptable and needs to be reconsidered. Do the authors have any evidence that historically there was not any other conversion of larger extent? Secondly, this section is pretentious and does not embed results in the context of a real discussion. Finally, Fig. 4 is out of place; it should be presented before the discussion.

The Section “Planted forest as a potential forest C sink” is disappointing. It is currently a jumble of possible explanations for the sink of planted forest while no mention is made of uncertainties.

The Section “Challenges and implications for the China’s planted forest” is probably the most relevant part to readers of Nature Communications. Nonetheless, it only addresses one Figure placed in the Supplement and does not relate to results in the main body of the manuscript.

The last sentence is indicative of the level of maturity of this manuscript. The authors engaged in a remarkable exercise of data collection, data crunching, map verification but did not convey the message of the actual value of their findings. The actual value comes from a production of results and a thoughtful interpretation of results, including those that are less convincing. This latter aspect was not deal with properly in the manuscript and I would see quite some efforts spent here to balance things out and provide a more objective perspective on the scientific results of this study.

Further to the presentation of errors, I find the role of uncertainties being underplayed in the overall evaluation of the results. Lines 556-569 state that most changes are improbable from a statistical point of view, which puts the question whether the magnitude of the changes and the pathways of conversion are correct. Indeed, I did not seem to find an external validation of these pathways in the manuscript (but just some total numbers) so it is hard to judge whether the numbers produced by this study are reliable. I truly hope that the authors can challenge my understanding here (see also comment below to lines 153-

156, Supplementary).

Would it be possible to add some insets of maps at full resolution in figure showing maps of planted forests for the whole of China?

Detailed comments

Line 45. How to access references 12, 13 and 14?

Line 74. "600,000 field samples". From methods and supplement I understood that these are not all "field" but several are photo-interpreted. Am I right?

Lines 71-77 and 79-86. These paragraphs are similar, consider merging.

Line 113. What does "total changed pixels" mean?

Line 117. What does "net increase area" mean?

Lines 119-120. What is meant with "indicating a decrease in overall net increasing planted forest area in most regions of China"?

Line 122. The comparison referring to Supplementary Fig. 1 is unclear. This figure illustrates the extent of the planting programs and areal extent per year. What is the source for the areal extent? National statistics or this study? Personally, I would rather appreciate at this stage a scatter plot that compares values considered as reference and values obtained in this study.

Line 161-162. I am intrigued by the conversion from natural forest to planted forest. What is the reason? This indeed is neither afforestation nor reforestation or am I wrong?

Line 170. What is the added value of Supplementary Fig. 10 when compared to Fig. 2a in this context?

Lines 174-178. What are multi-change events crop-to-forest?

Line 187. What does “C storage leading by planted forest area expansion” mean?

Line 191. Are 8 TgC/year the result of 240 TgC / 30 years? This is rather simplistic as it assumes a linear rate which I did not see in the manuscript. In any case, I would appreciate a confidence interval on the total.

Line 192. 11% is with respect to all forests in China (natural+planted)?

Line 195. What is 99% referring to?

Lines 196-198. The sentence needs to be revised. What do “obviously increase”, “largest three net accumulation” and “declined area” mean?

Line 203 (and then again line 601). The statement “canopy height relates linearly with C density across large scales” is indefinite and partly incorrect. The reference to Fang et al., 2006, clearly shows a non-linear trend from short to tall forest (Fig. 1 therein). A linear trend characterized forests taller than 10 m, which may not be the case for newly planted forest as those addressed in this manuscript.

Line 205 (and then again line 606). Claiming that P-PET is the best predictor for forest canopy height is challenged by the plot in Figure 3. The best predictor would be spaceborne LiDAR interpolated with some kind of spatial predictor in my opinion, or am I missing something?

Line 220, Fig. 3. What is the spatial scale, i.e., spatial resolution, of the P-PET data layers and the canopy height? Are these comparable?

Line 223, caption to Figure 3b, should refer to canopy height rather than P-PET.

Lines 225-229. Replace links with footnotes or cite elsewhere.

Line 266. Why until 2014?

Line 351. "mean values" of what?

Line 354. "change events" are by definition "dynamic". Please revise wording here.

Lines 473-490 (Field samples) and lines 492-504 (Landsat data) should be switched.

Line 474. Of the 600,000 samples, only about 180,000 were explained. What is the origin of the remaining 420,000 samples?

Line 475. What are "multi-source ways"?

Line 474-475. If the samples were collected after 2003, which data were used to train the classifier to derive maps for the time period 1990-1995-2000?

Line 477. How were the crowdsourced samples used? They were collected since 2018 so in theory useful only to support maps for 2020.

Line 484. Please clarify here and throughout the whole text including the Supplement whether the forest masks were annual or quinquennial.

Line 486. What does "target regions" mean?

Line 488. How were "significant changes" defined?

Line 502-504. Do I understand correctly that the mosaics are not gap-free? If so what is the size of the gaps? Is this relevant at all?

Line 508. Which "scales" are you talking about?

Line 509-510. Why would you compare a map based on data for 2004-2008 with your

planted forest maps representative of the epochs 2000 and 2010?

Line 513. “temporal features” of what?

Line 510. “Surface reflectance data” is it the Landsat-based?

Line 523 “Median value” of what?

Line 529. It is referred to Supplementary Note 2, which however repeats what has been reported in this paragraph.

Line 544. Of the “four” validation methods, the two last ones are just a comparison. Please rephrase.

Line 553. “agreed well” should be replaced with a more precise statement.

Line 594. Could you please elucidate on the link between vegetation type map and the maps of planted forest.

Line 620. Does not Nature Communication have a policy that all data material should be open and public? If not, which “data” will be made available. I think it is responsible act of transparent science to release maps and accounts.

Supplementary Notes

Lines 20-21. How could the maps of 1990, 1995 and 2000 be assessed if no data were available before 2003?

Line 21. Figure 23 shows overall accuracies but the real interest here is on a confusion matrix listing all classes that were mapped.

Line 43. Please provide a reference for the harmonic analysis.

Line 61. What is an “original website”?

Line 74. What does “exported overall accuracies” mean?

Line 91 and line 114. “validation data from stratified random sampling” is unclear. Please specify instead the origin of the data

Line 94. “maps and datasets from other studies” is generic. Please specify.

Line 106-111. I am not sure that Supplementary Fig. 12b and Fig. 14 are correctly referred to here.

Lines 111-112. Can you please explain why accuracy increased with time? Is it a consequence of the higher quality of the Landsat data?

Lines 117-120. I am puzzled by the “random generation of points” in a “grid with resolution of 0.1 x 0.1 deg”. Could you please clarify?

Lines 122-126. Where are the accuracies shown?

Line 130. Are the statistical data published on a yearly basis? What about the period 1990-2000?

Line 132. The large dispersion of data points in Supplementary Fig. 18 should be addressed as well.

Line 142. What exactly was digitized here?

Line 143. What is Supplementary Fig. 19 supposed to show? As far as I can judge, the figure does not show agreement. It is a comparison of totals. Both maps could be wrong in how

planted forests are distributed but the totals agree. Any thoughts on how to address this issue?

Lines 153-156. Claiming that an uncertainty < 0.4 is low-to-moderate (rather than “medium”) whereas it is high when > 0.4 is misleading. It is generally agreed upon that high accuracy is > 0.7 . If this understanding is correct, the planted forest maps are highly uncertain, with some relevant impact on the relevance and the outcome of this study. Please clarify.

Line 308. “Samples” are difficult to discern when the figure is printed on paper.

Line 322. What is the reason for adding regression lines here? These lines indicated that the slope and the intercept increase with time. Could it be a symptom that the maps suffer from the quality of the Landsat data? In principle, forest patches of a certain size should always be mapped at a certain size \pm some uncertainty whereas here we observe a clear temporal trend.

Line 326. What is shown in these maps?

Reviewer #2 (Remarks to the Author):

This could be a data paper, but it falls short of a research paper targeting carbon stock assessment. The author used Landsat TM images and forest samples to generate planted forest maps in China from 1990 to 2020. Despite the fact that I believe the planted forest maps do add new knowledge to our understanding, the focus of this study - carbon stock assessment - was too simplified to be convincing.

First, the carbon density of each vegetation type was obtained from the study of Lai et al (2016) (note that it is not Li et al, I believe). This inherits the uncertainties from the study (Lai's study used mean values for the same vegetation type compiled from different studies). Besides, forest types are not fixed over time. Instead, they evolve and change during succession. Using a fixed type to extrapolate the carbon potential will be biased.

Second, using tree height as a proxy to project carbon density is also problematic. Indeed, there is a close relationship between tree height and P-PET. However, tree height is determined by site index, and factors such as soil fertility, slope, and aspect are important. Moreover, height is also related to the planting density of the trees. Using a simplified relationship of height and P-PET to further extrapolate the carbon stock potential is highly unreliable.

Because of the above-mentioned problems, I highly doubt the reliability of the discussion and conclusions. For example, in lines 322-325, this is not reasonable as there are no such absolute differences between planted and natural forests. An instance is that human interventions to expedite the succession process via complementary planting of seedlings, selective cutting, and pruning in natural secondary forests will also help increase carbon stock. Are those forests natural or planted?

Reviewer #3 (Remarks to the Author):

Comment for authors :

This manuscript obtained planted forest maps for each five-year interval from 1990 to 2020 based on Landsat data and random forest algorithm, and analyzed the sources of planted forest expansion in China. Meanwhile, this study quantified the C storage resulting from the expansion of planted forest over the past 30 years, and evaluate the biomass C storage potential of the additional planted forest. Overall, this study is truly unprecedented, carrying milestone significance in enhancing our understanding of the temporal and spatial changes within China's planted forests and their pivotal role in mitigating global climate change. However, there are some minor issues that need to be solved before publication.

Main Comments:

- 1、 The manuscript mentions that canopy height is a significant factor related to forest carbon storage potential. However, it appears that the author did not take canopy height into account when calculating the carbon storage conversion from other land types to

planted forests. Instead, the author calculated carbon storage solely based on the conversion area. This oversight raises questions about the accuracy of the carbon storage estimate.

2、 In light of the points mentioned above, we assert that the results pertaining to P-PET in the final section (Lines 202-219) have limited relevance to the overall content. Their removal will not disrupt the paper's logical structure but will enhance its conciseness.

3、 In the analysis of Land Use and Land Cover (LULC) transfer, the author identifies shrubland, grassland, and natural forest as the primary sources of plantation forest. While the author provides comprehensive explanations and references for shrubland and grassland in the discussion, there appears to be a gap in addressing the key issue of natural forest loss, which is arguably the most critical forest type to protect. It is evident that a more in-depth discussion regarding the reasons behind the loss of natural forests is warranted.

4、 The quality and consistency of the two vegetation-type maps are crucial factors influencing the reliability of the results concerning carbon storage changes. I have concerns regarding the consistency in classifying vegetation types within these two maps. If different classification systems were employed, it would render them non-comparable. Therefore, it is imperative that the author provides more detailed information about these two vegetation maps.

Minor Comments:

1、 Line 290, native forest should be natural forest;

2、 Line 304 underscores should be underscore.

3、 The contents of subgraph b, and d of Fig 3 are not corresponding with the title of the subgraph.

4、 Line 541, it is suggested to add respectively at the end of the sentence to increase readability.

5、 Line 502, the full name of the FMASK algorithm should be given at the first time.

6、 Line 539 to 541, the classification model adapted to each region and period should be provided.

- 7、 Line 557, work should be works.
- 8、 Line 587, delete word code.
- 9、 Line 589, vegetation-type maps should be vegetation-type map.

Response to Reviewers

Reviewer #1:

This manuscript addressed the quantification of areal extent and carbon storage of planted forests in China from 1990 to 2020. The detection of planted area is achieved with Landsat images. Classification and validation of the newly planted forest maps are supported by an extensive dataset of sample measurements and auxiliary datasets. Dynamics of land cover are analyzed and attributed to afforestation and reforestation programs. The planted forest maps are then related to measurements from the National Forest Inventory to quantify their carbon storage and their potential uptake of carbon.

The manuscript is in principle of interest to readers of Nature Communications because it provides a detailed explanation of land cover trajectories in China during the last 30 years. It is acknowledged that this is a massive study, supported by many figures and tables in the Supplement. It is a strenuous task to bring all the evidence together in a single manuscript. However, I find the manuscript overloaded with topics, with a tendency towards the remote sensing component, which in turn makes the manuscript less appealing to this journal in my opinion. Aspects related to the implications of the findings are less pronounced, giving the feeling that the manuscript has not reached its natural balance yet.

Based on my background on remote sensing, I also see several aspects that need to be clarified concerning the derivation of the forest planted maps, their accuracy and the quantification of the carbon storage. To appreciate whether the numbers provided are reasonable, a thorough analysis of errors is needed in addition.

The authors find below a set of general comments that may stimulate some thinking on how the manuscript shall be improved and a more detailed list of comments having the aim of polishing the text. The authors are strongly encouraging to revise their manuscript and submit a new version to Nature Communications.

Response: Thank you for your valuable and positive feedback. Your insightful comments have been instrumental in enhancing the quality of our work. We have carefully considered each of your comments and made substantial revisions to the manuscript. Below, we provide comprehensive point-by-point responses to address the specific issues raised during the review process.

General comments

1 The aim apparently was to monitor the carbon storage dynamics. This aspect is vaguely addressed as it is not clear how C was validated and what is the magnitude of the error bars (text on lines 349-357 is not satisfactory).

Response: We agree with the comment that the reviewer pointed out. We have carefully addressed these issues in our revised manuscript.

First, our primary aim in this study was indeed to monitor the spatiotemporal dynamics of carbon storage in China's planted forests. To address this aspect more explicitly, we have extensively revised the introduction section of the paper to emphasize the core objectives of our study. We kindly invite you to review the modified introduction in our revised manuscript in Line 40-50 for a more precise presentation of the research theme.

Second, we have provided detailed information on the validation of carbon storage estimates using field plot (Xu et al., 2019) and Yang's biomass map (Yang et al., 2023), suggesting an R^2 of 0.6 to 0.7 (Fig. R1). Based on forest inventory data, we re-calculated the carbon density for different vegetation types (Methods Line 486-541, Supplementary Table 6), and described magnitude of error for carbon storage estimates (Fig. R2 and Fig. R3). These error calculations have been rigorously conducted through robust statistical analysis to ensure their reliability. Based on the new results, we analyzed the carbon storage change of planted forest from 1990 to 2020 (Line 143-248).

Furthermore, we have delved into an in-depth analysis of the uncertainties about carbon storage estimation. We have discussed potential sources of uncertainties and highlighted the proactive measures taken in our study to minimize these uncertainties (Line 549-614). We believe that these modifications and additions have significantly improved the clarity, credibility, and completeness of our research.

[figure redacted]

Fig. R1. Comparison with Yang's biomass dataset (a) and field survey(b).

Fig. R2 Changes of C in planted forest biomass over national and regional scales.

Fig. R3. C storage changes during various periods at different regions.

2 The mapping sequence is not entirely clear. This is my understanding:

- 1) A forest mask for each epoch was created from existing land cover maps. Question here is how was “forest” defined. Could it be that the definition behind each dataset was different, leading to different accuracy statistics which then influenced the selection of the reference for the forest mask?

Response: The used three land cover dataset (GLC_FCS30 generate by Zhang et al (2021), CLCD generated by Yang and Huang et al (2021), and CLUD generate by Liu et al (2014)) had different forest definitions. We acknowledge that this difference in forest definition could influence accuracy statistics and, consequently, the reference selection for the forest mask. In our study, based on a straightforward accuracy comparison, we selected the annual China Land Cover Dataset (CLCD) (Yang et al., 2021) as the foundational dataset.

It is essential to note that our validation samples were entirely independent and collected based on field samples. These samples faithfully represent on-ground situations and provide an accurate reflection of the actual land cover types. Thus, we leveraged these independently collected samples to assess the accuracy of each forest mask and determine which definition aligns better with the China's forest distribution. As a result, the dataset we selection is rational.

We appreciate the reviewer's attention to this matter and hope that this explanation clarifies this concern.

2) Planted forest and natural forest maps (yearly? for every five years?) were obtained from Landsat metrics + auxiliary datasets in correspondence of the forest mask. When evaluating these maps, user/producer accuracies are provided for the two classes but the manuscript discusses conversions from shrubland, cropland and other classes to/from natural and planted forest. How where the non-forest classes mapped? What is the accuracy statistics for these classes?

A flowchart would very much help.

Response: We generated planted and natural forest maps for the years of 1990, 1995, 2000, 2005, 2010, 2015, 2020 based on correspondent forest masks of CLCD. We did not identify other land cover types such as cropland, grassland, shrubland, etc. Instead, we directly used the land cover data from CLCD for the analysis of conversion in cropland, grassland, shrubland, etc. We added a flowchart (Fig. R4) to describe the planted forest mapping procession, mainly include selecting forest mask selection, generate training and testing sample for different periods, and mapping.

Fig.R4. Flowchart of mapping planted forests.

3 Several comparisons are mentioned throughout the manuscript but no figure compares values (see also detailed comment for line 122). The number of figures could actually be reduced if reference data and mapped data were co-plotted. This would avoid jumping back and forth in the manuscript and let the reader appreciate how well the mapped data corresponds to the reference.

Response: We are very grateful for the reviewer's suggestions. Following the reviewer's comments and the newly revised manuscript, we have corrected the figures and tables, particularly those in the manuscript file. The figures in the Supplementary have also been modified. Please check these revisions.

4 The Section on "Present and future planted C forest storage potential related to water availability" has certain merits but seems to be out of scope in this manuscript and should be reconsidered. I would focus on a transparent presentation of the planted forest maps (strengths and weaknesses), and reinforce the text describing the C balance of planted forests with implications in terms of policy and future developments with respect to future forest expansion programs (if any of course).

Response: We highly agree with the reviewer's suggestion. After integrating the comments from you and other reviewers, the "Present and future planted C forest storage potential related to water availability" section has been removed from manuscript. In line with the reviewer's suggestions, we focused on carbon storage

changes resulting from spatiotemporal dynamic of China's planted forests. Therefore, we have added more results and discussions regarding carbon storage estimates, changes, and uncertainty analysis in revised manuscript. Please refer to the response to first General Comment.

5 The title of the Section "The largest LULC conversion in human history" is not acceptable and needs to be reconsidered. Do the authors have any evidence that historically there was not any other conversion of larger extent? Secondly, this section is pretentious and does not embed results in the context of a real discussion. Finally, Fig. 4 is out of place; it should be presented before the discussion.

Response: We agree and have deleted it. Meanwhile, we re-organized the discussion section according to the updated results. Please see the revised manuscript in Line 255-269.

6 The Section "Planted forest as a potential forest C sink" is disappointing. It is currently a jumble of possible explanations for the sink of planted forest while no mention is made of uncertainties.

Response: Agree, please see the response to fourth General Comment.

7 The Section "Challenges and implications for the China's planted forest" is probably the most relevant part to readers of Nature Communications. Nonetheless, it only addresses one Figure placed in the Supplement and does not relate to results in the main body of the manuscript.

Response: Following your suggestion, we re-organized the discussion section, and strengthened the discussion about the results. Please see Discussion Section in Line 249-356.

8 The last sentence is indicative of the level of maturity of this manuscript. The authors engaged in a remarkable exercise of data collection, data crunching, map verification but did not convey the message of the actual value of their findings. The actual value comes from a production of results and a thoughtful interpretation of results, including those that are less convincing. This latter aspect was not deal with properly in the manuscript and I would see quite some efforts spent here to balance things out and provide a more objective perspective on the scientific results of this study.

Response: Thank you for the professional suggestions. As mentioned in response above, the revised manuscript focused on the spatiotemporal dynamics of carbon

storage in China's planted forests. In Discussion Section, we have primarily discussed the reasons for the spatiotemporal dynamic of China's planted forests, and the resultant increase in planted forest carbon storage from the planted forest expansion in area and its growth, as well as the uncertainties associated with planted forest maps and results of carbon storage. Finally, we concluded "Our study is the first to characterize the spatiotemporal dynamics of C storage in the biomass of China's planted forests over a thirty-year period, from 1990 to 2020. We found that the increase in C storage within China's planted forests can be primarily attributed to the expansion of planted forest areas. This expansion has largely occurred through the conversion of croplands, shrublands, and grasslands into planted forest areas. In addition to area expansion, the growth of China's planted forests themselves represents another significant factor contributing to C storage. Post-2000, a policy-driven push for expanding planted forest areas has led to rapid increase, with the majority of these planted forests currently in their young or middle age. Based on these results, we anticipate that the newly expanded planted forests will possess substantial C storage potential in the future, potentially playing a key role in helping China meet its Carbon Neutrality Target by 2060.". Please see Discussion Section in Line 358-369.

9 Further to the presentation of errors, I find the role of uncertainties being underplayed in the overall evaluation of the results. Lines 556-569 state that most changes are improbable from a statistical point of view, which puts the question whether the magnitude of the changes and the pathways of conversion are correct. Indeed, I did not seem to find an external validation of these pathways in the manuscript (but just some total numbers) so it is hard to judge whether the numbers produced by this study are reliable. I truly hope that the authors can challenge my understanding here (see also comment below to lines 153-156, Supplementary).

Response: Thanks for pointing this out. We supplemented more details about the error and uncertainty analysis. First, we recalculated the carbon density for various vegetation type, and their standard deviations (Methods about planted forest carbon storage in Line 486-541, and Supplementary Table 6), and added the error range to analysis the results of carbon storage dynamics step by step (Results section of line 143-248). Second, we validated results of the carbon storage estimation using field survey and existing remote sensing product, indicating an R^2 of 0.6 to 0.7 (Fig R1).

Third, we conducted an in-depth analysis about the source of uncertainties to results of the carbon storage estimation (Line 543-614).

10 Would it be possible to add some insets of maps at full resolution in figure showing maps of planted forests for the whole of China?

Response: Following your suggestion, we added planted forest maps with 30m resolution in Supplementary Fig.1 as shown in Fig.R5.

Fig.R5. Planted forest maps from 1990-2020

Detailed comments

11 Line 45. How to access references 12, 13 and 14?

Response: These three references are Chinese government reports, if you are interested in these three reports, please contact the corresponding author.

12 Line 74. “600,000 field samples”. From methods and supplement I understood that these are not all “field” but several are photo-interpreted. Am I right?

Response: Yes, the crowdsourced samples were photo-interpreted by vegetation ecology experts.

13 Lines 71-77 and 79-86. These paragraphs are similar, consider merging.

Response: Thanks. We merged these two sentences, please check in line 78-79 of Main Text.

14 Line 113. What does “total changed pixels” mean?

Response: We conducted a spatiotemporal analysis of planted forest area changes at a 0.1° grid scale. Here, "total changed pixels" refers to all the grid cells where changes in plantation forest area have occurred. To avoid misunderstanding, we revised this to “changed pixels”. Please see line 119-120, and 123.

15 Line 117. What does “net increase area” mean?

Response: "Net increase area" refers to the increase in planted forest area over a period. For example, it is calculated by subtracting the planted forest area in 1990 from the planted forest area in 1995, representing the net increase in area over those five years.

16 Lines 119-120. What is meant with “indicating a decrease in overall net increasing planted forest area in most regions of China”?

Response: We conducted trend analysis using linear regression within the 0.1° grid cells for the increase in planted forest area every five years. This means that the results indicate that in most grid cells, there is a decreasing trend in the increase of planted forest area every five years.

17 Line 122. The comparison referring to Supplementary Fig. 1 is unclear. This figure illustrates the extent of the planting programs and areal extent per year. What is the source for the areal extent? National statistics or this study? Personally, I would rather appreciate at this stage a scatter plot that compares values considered as reference and values obtained in this study.

Response: The data in Supplementary Fig. 1 is sourced from national forest statistics.

We have compared the plantation forest area obtained in our study with national statistical data from 2000 to 2020, as shown in the following Fig.R6. Our data shows a good level of consistency with the statistical data, with an R^2 ranging between 0.76 and 0.87.

Fig. R6. Comparison with statistical planted forest area.

18 Line 161-162. I am intrigued by the conversion from natural forest to planted forest. What is the reason? This indeed is neither afforestation nor reforestation or am I wrong?

Response: This is a highly intriguing and worth addressing question. In China, the transformation of natural forests into planted forests is primarily influenced by a dual impact of policy-driven and economic factors. Since the 1970s and 1980s, China has seen extensive deforestation of natural forests to meet the demands of national economic development, resulting in a decline in the quality of natural forests (Cao et al., 2011). In response to the deteriorating state of natural forest ecosystems, the Chinese government has successively introduced national and provincial-level policies for the protection of natural forests (Chen et al., 0222). These policies promote afforestation to enhance the structure of natural forests, leading to the gradual conversion of some lower-quality natural forests into planted forests (Cao et al. 2011). Economic development requires a supply of timber and related products. To

meet these economic development needs, particularly in the southeastern tropical forest regions of China, there are many cases of natural forests being converted into Chinese fir planted forests (Yang et al. 2019). In addition, the establishment of fast-growing and high-yield planted forests for economic purposes has also led to the loss of natural forests (Zhai et al. 2017). However, since 2015, China has ceased comprehensive commercial logging, and the area of natural forests has slowly increased, as evidenced by national statistical data.

19 Line 170. What is the added value of Supplementary Fig. 10 when compared to Fig. 2a in this context?

Response: Supplementary Fig. 10 illustrated the predominant types of land cover conversion in different regions for change events (single and multiple events), whereas Fig. 2a present the primary land cover types converted into plantation forests from 1990 to 2020, which differs from the information in Supplementary Fig. 10. We have revised the order of figures in manuscript, the Fig.2a has been adjusted to Fig. 4a. Please check in Line 238-244.

20 Lines 174-178. What are multi-change events crop-to-forest?

Response: To interpret this transformation, it is necessary to consider China's afforestation policies. Firstly, the GFG has been a significant factor driving the expansion of China's planted forests. However, after afforestation, some regions may experience improper management of planted forests, leading to their mortality and necessitating reforestation, resulting in interactive changes. Research indicated that from 1952 to 2005, the overall survival rate of trees in afforestation projects was only 24% (Wang et al., 2007), with a survival rate of only 15% for the Three-North Shelterbelt Program (Li, 2001, Cao, 2008). This has resulted in multiple rounds of land cover type changes between planted forest and other land use types. China has innovated afforestation techniques, such as runoff forestry and deep planting, which integrate seedling and container-based methods, are actively being pursued to enhance the drought resistance and survival rates of planted forests (State Forestry and Grassland Administration of China, 2019).

21 Line 187. What does “C storage leading by planted forest area expansion” mean?

Response: This subheading was intended to emphasize the contribution of the expansion of China's planted forest area over the past three decades to the increase in

carbon storage in planted forests. To reduce ambiguity, we have deleted this subheading.

22 191. Are 8 TgC/year the result of 240 TgC / 30 years? This is rather simplistic as it assumes a linear rate which I did not see in the manuscript. In any case, I would appreciate a confidence interval on the total.

Response: Thank you for the useful suggestions which we considered carefully. To obtain a more precise estimate of the changes in carbon storage in China's planted forests, we have conducted a comprehensive effort about China's planted forest carbon storage estimates. Please refer to the response to first General Comment. Please check in Line 143-248.

23 192. 11% is with respect to all forests in China (natural+planted)?

Response: We have revised this section. Please see detailed in 143-185.

24 Line 195. What is 99% referring to?

Response: In the previous manuscript version, the aim was to express that carbon storage from the conversion of cropland, shrubland, grassland, and natural forests into planted forests constituted 99% of the total increase in carbon storage in planted forests. As we have conducted a reevaluation and provided error estimates, these results have consequently been revised. Please check in line 186-244.

25 196-198. The sentence needs to be revised. What do “obviously increase”, “largest three net accumulation” and “declined area” mean?

Response: Thanks for your suggestion. We have corrected it and please in line 186-244.

26 Line 203 (and then again line 601). The statement “canopy height relates linearly with C density across large scales” is indefinite and partly incorrect. The reference to Fang et al., 2006, clearly shows a non-linear trend from short to tall forest (Fig. 1 therein). A linear trend characterized forests taller than 10 m, which may not be the case for newly planted forest as those addressed in this manuscript.

Response: We highly agree with the reviewer's suggestion and thank you for pointing out this issue. We only discussed this issue in line 321-325, which focused on the favorable climatic contributing to the growth and survival rate of planted forest. Please also refer to the response to first and fourth General Comment.

27 Line 205 (and then again line 606). Claiming that P-PET is the best predictor for forest canopy height is challenged by the plot in Figure 3. The best predictor would be spaceborne LiDAR interpolated with some kind of spatial predictor in my opinion, or am I missing something?

Response: Thank you again. Please refer to the response to first and fourth General Comment, and the last response to line 238.

28 Line 220, Fig. 3. What is the spatial scale, i.e., spatial resolution, of the P-PET data layers and the canopy height? Are these comparable?

Response: We analyzed the P-PET and carbon storage within the same 0.1° grid.

29 Line 223, caption to Figure 3b, should refer to canopy height rather than P-PET.

Response: We agree. However, we deleted this figure according the comments of all reviewers. Please see the response to General Comment 1, 4 and response to line 238 and 240.

30 Lines 225-229. Replace links with footnotes or cite elsewhere.

Response: This figure has been deleted according to the comments from you and other reviewers.

31 Line 266. Why until 2014?

Response: There is no publicly accessible data after 2014.

32 Line 351. “mean values” of what?

Response: Mean values of carbon density for the same vegetation type.

33 Line 354. “change events” are by definition “dynamic”. Please revise wording here.

Response: Thanks for the suggestion. We have revised the content and defined in line 478-484.

34 Lines 473-490 (Field samples) and lines 492-504 (Landsat data) should be switched.

Response: Thanks for the suggestion, we switched these two sections. Please see line 381-416.

35 Line 474. Of the 600,000 samples, only about 180,000 were explained. What is the origin of the remaining 420,000 samples?

Response: I am sorry for the ambiguity. Among the 600,000 field samples, not only

forest samples are included, but also other vegetation type survey data, such as grassland, shrubland etc. We only focused on forests, so we used 180,000 field samples related to forests

36 Line 475. What are “multi-source ways”?

Response: This research acquired more than 600,000 vegetation survey datasets through crowdsourcing, field surveys, and national forest inventories. To avoid ambiguity, we revised the line 397-398.

37 Line 474-475. If the samples were collected after 2003, which data were used to train the classifier to derive maps for the time period 1990-1995-2000?

Response: Utilizing the yearly CLCD forest mask data and NDVI temporal series from 1990 through 2020, we performed a change analysis on each sample. Samples that exhibited no change were chosen as the reference samples for identifying planted forests for each period, from which we derived our training and verification samples for every interval. The process of constructing samples for various periods has been added; for details, see lines 394-416 in the revised manuscript.

38 Line 477. How were the crowdsourced samples used? They were collected since 2018 so in theory useful only to support maps for 2020.

Response: Similar to response to Comment 37. By conducting a time-series change analysis on crowdsourced samples, those without changes are used as the reference dataset for each period, further divided into training and validation samples. We have added methodological details, and please refer to lines 394-416.

39 Line 484. Please clarify here and throughout the whole text including the Supplement whether the forest masks were annual or quinquennial.

Response: We appreciate the recommendation. Explanatory notes have been incorporated throughout the revised manuscript and supplement to clarify that the forest masks (CLCD) pertain to each year from 1990 to 2020. Please see line 404-405.

40 Line 486. What does “target regions” mean?

Response: They are the forest regions without changes. We revised it and please see line 394-416.

41 Line 488. How were “significant changes” defined?

Response: Here, referencing the research of Khunrattanasiri et al. (2022), we determine whether each sample has undergone changes based on whether the

minimum value of the NDVI time series for each sample is less than 0.6. To prevent misinterpretation, modifications and clarifications have been made in the manuscript; please refer to lines 410-412.

42 Line 502-504. Do I understand correctly that the mosaics are not gap-free? If so what is the size of the gaps? Is this relevant at all?

Response: We computed the quantity of null values present in the five-year composite imagery, and the findings indicated that post-cloud processing, null-value pixels in forest regions between 1990 and 2020 accounted for merely 0.1% to 0.2% (Fig. R7), signifying a minimal effect. In our discussion of uncertainties, we have augmented the section to address the impacts arising from image quality and missing data; see lines 549-557 for this detail.

Fig. R7. Landsat gaps for various periods

43 Line 508. Which “scales” are you talking about?

Response: The “scales” mean national and provincial scales, we revised to “national and provincial scales” in Line 455.

44 Line 509-510. Why would you compare a map based on data for 2004-2008 with your planted forest maps representative of the epochs 2000 and 2010?

Response: This map, dating from 2004 to 2008, was digitized using data from a comprehensive survey of forests in China. It is currently the only dataset that accurately reflects the distribution of China's planted forests. This dataset has been widely utilized in various research studies, such as the work conducted by Peng and others. To facilitate a meaningful comparison with this dataset, we selected the generated planted forest maps in 2000, 2005, and 2010 to compare with this dataset, because they are close in year.

45 Line 513. “temporal features” of what?

Response: The term "temporal characteristics" refers to the features associated with the time series changes of vegetation indices. In this study, we primarily utilized metrics such as amplitude and phase to analyze these temporal characteristics. See details in Supplementary Note 2.

46 Line 510. “Surface reflectance data” is it the Landsat-based?

Response: Yes, it is from Landsat.

47 Line 523 “Median value” of what?

Response: "Median value" refers to the median value of multiple images on the same pixel.

48 Line 529. It is referred to Supplementary Note 2, which however repeats what has been reported in this paragraph.

Response: Agree and we deleted the repeated content.

49 Line 544. Of the “four” validation methods, the two last ones are just a comparison. Please rephrase.

Response: Thanks for your patient correction. We rephrased this sentence, please see Line 415.

50 Line 553. “agreed well” should be replaced with a more precise statement.

Response: We revised this to “exhibited a high degree of concordance”, please line 4461.

51 Line 594. Could you please elucidate on the link between vegetation type map and the maps of planted forest.

Response: The vegetation type map is a geographic representation that classifies and

illustrates the various types of vegetation. It contains 17 forest types, primarily representing a potential spatial distribution of these types (Zhang 2007). Based on this vegetation map and spatial nearest union method, we can divide the planted forest map into different forest types. Then according to the study of Lai et al. (2016), using carbon density, we can estimate carbon storage of planted forests.

52 Line 620. Does not Nature Communication have a policy that all data material should be open and public? If not, which “data” will be made available. I think it is responsible act of transparent science to release maps and accounts.

Response: The data will be made available following the article's publication.

Supplementary Notes

53 Lines 20-21. How could the maps of 1990, 1995 and 2000 be assessed if no data were available before 2003?

Response: Please see the response to Comment 37, and see details in line 359-381.

54 Line 21. Figure 23 shows overall accuracies but the real interest here is on a confusion matrix listing all classes that were mapped.

Response: We have solely classified planted and natural forests on the basis of the CLCD forest masks; for other types, we utilized the existing CLCD products. Please also the response to the General comments 2-2).

55 Line 43. Please provide a reference for the harmonic analysis.

Response: Thanks for the suggestion, we added the study of Zhang et al. (2015) as a reference. Please see Supplementary line 46

56 Line 61. What is an “original website”?

Response: To avoid misunderstanding, we deleted the word of “original”, please see line 62.

57 Line 74. What does “exported overall accuracies” mean?

Response: we determined the parameter according to the overall classification accuracies, which were calculated using the testing samples. We revised this sentence, please see Supplementary line 76-77.

58 Line 91 and line 114. “validation data from stratified random sampling” is unclear. Please specify instead the origin of the data

Response: Initially, we have to concede that a clerical error occurred; the samples were not derived from stratified random sampling but from systematic sampling,

which we randomly constituted. As the field validation samples utilized were not randomly selected, this may impart bias to the validation results (Olofsson et al., 2014); hence, systematic sampling was adopted to enhance the precision verification of the data product (Rizayeva et al., 2023). Our initial step was to create a 0.1° grid covering the entire country, followed by the establishment of a central point within each grid cell, combining these central points with forest survey maps and high-resolution remote sensing imagery from Google Earth (Wang et al. 2021), with forest types ascertained by experts in vegetation ecology, subsequently utilizing these systematically collected samples to validate these maps. For additional details, consult the Supplementary material, lines 118-131.

59 Line 94. “maps and datasets from other studies” is generic. Please specify.

Response: Thanks for the suggestion. We revised this sentence to “intercomparison with forest inventory map”. Please see line 97-98.

60 Line 106-111. I am not sure that Supplementary Fig. 12b and Fig. 14 are correctly referred to here.

Response: This is a misquote, we revised, and it should be Supplementary Fig. 14b and Fig. 16a. In the revised manuscript, they have been listed as Supplementary Fig. 4b and 11b.

61 Lines 111-112. Can you please explain why accuracy increased with time? Is it a consequence of the higher quality of the Landsat data?

Response: We attribute this to two main factors: first, the quality of remote sensing imagery. It is well recognized that the quality of Landsat imagery has been progressively improving, which in turn directly impacts the precision of classification (Yang and Huang 2021). Second, it pertains to the training and validation samples employed, as our samples span from 2003 to 2020, predominantly amassed post-2015, and when we extrapolate the training and validation samples for each period using NDVI time series change detection techniques, unavoidable errors in the sample set construction arise from the inherent inaccuracies of change detection, the earlier the time frame of the sample extrapolation (nearing 1990), the more substantial the resultant noise, which ultimately impairs the classification accuracy (Calderón-Loor, Hadjikakou, and Bryan 2021).

62 Lines 117-120. I am puzzled by the “random generation of points” in a “grid with resolution of 0.1 x 0.1 deg”. Could you please clarify?

Response: Please see response to Comment 58.

63 Lines 122-126. Where are the accuracies shown?

Response: There is a missing reference, the accuracies were shown Supplementary Fig.3. We added this in Line 138 and 141.

64 Line 130. Are the statistical data published on a yearly basis? What about the period 1990-2000?

Response: According to the statistical year book of China, the planted forest data were only available from 2000 to 2020, and updated every five years.

65 Line 132. The large dispersion of data points in Supplementary Fig. 18 should be addressed as well.

Response: Thanks for the suggestion. We re-drawn this figure, as shown in Fig.R6. Please see response to Comment 17.

66 Line 142. What exactly was digitized here?

Response: The planted forest map of National Forest Inventory Map was digitized (Peng et al. 2014).

67 Line 143. What is Supplementary Fig. 19 supposed to show? As far as I can judge, the figure does not show agreement. It is a comparison of totals. Both maps could be wrong in how planted forests are distributed but the totals agree. Any thoughts on how to address this issue?

Response: Supplementary Figure 19 illustrates the disparity within a 0.1° grid between the planted forest area derived from our study and that from the digitized National Forest Inventory Map, which is obtained during 2004 to 2008 conducted by State Forestry and Grassland Administration of China. Hence, we compared our results in 2000, 2005 and 2010 with this inventory map because they're close in time. As shown in Fig.R8, panels *a-c* displays the area differences within 0.1° grids between the mapped results for 2000, 2005, and 2010 and the National Forest Inventory Map. Panel *d* presents the frequency distribution based on the area difference range, with numbers indicating proportions. It can be observed that for the years 2000-2010, the grids with area differences less than 30 square kilometers account for 79.3%, 79.1%, and 72.7%, respectively, indicating good consistency between the forest map generated in this study and the inventory data. To provide a clearer presentation, we have added explanations for panels a-d in the figure captions, and please see line 171-174.

We acknowledge that there may be errors in both the National Forest Inventory Map and the maps we generated. However, the National Forest Inventory Map is currently the most authoritative dataset for representing the distribution of China's planted forests. Many scholars have used it to study the ecological benefits of China's planted forests, such as Peng et al. (2014), who utilized it to assess the climatic benefits of afforestation in China. Additionally, the spatial database of planted forest released by the World Resources Institute also uses this data as a representation of China's planted forests (Harris et al., 2019). Therefore, to validate the accuracy of the map produced in this study, we are making the assumption that this dataset is accurate for the purpose of comparison.

Fig.R8. Comparison of planted forest area within each 0.1° between the resultant maps and available planted forest dataset in 2000, 2010 and 2020. a, b and c are the spatial distribution of area difference; d is the frequency statistics of grid counts with various area differences.

68 Lines 153-156. Claiming that an uncertainty < 0.4 is low-to-moderate (rather than “medium”) whereas it is high when > 0.4 is misleading. It is generally agreed upon that high accuracy is > 0.7. If this understanding is correct, the planted forest maps are highly uncertain, with some relevant impact on the relevance and the outcome of this study. Please clarify.

Response: The uncertainty of each pixel in the resultant maps was assessed using the

outputs of the RF classifier. Classification uncertainty for a pixel, labeled "u," is conveyed through a probability vector produced by the RF probabilistic classifier (Loosvelt et al., 2012). This vector contains probabilities corresponding to the pixel's assignment to various classes. The maximum probability in the vector, termed " p_{max} ", indicates the most probable class based on the RF model. The value "u", calculated as $1 - p_{max}$, gauges the strength of class assignment and the potential for confusion. A high u value suggests ambiguity or confusion between classes, while a low u value implies greater model confidence in the assigned class.

Therefore, the uncertainty <0.4 means that the classification probabilities are greater than 60%, which accounts for over 80% of all classification region (Liu et al., 2004). We recalculated the proportion of the uncertainty less than 0.3, which was 77.78%, 77.70%, 80.06%, 78.52%, 74.60%, 72.49%, and 65.41%, respectively from 1990 to 2020. This implies that, except for the year 2020, during all other periods, there is over 70% planted forest pixels having a recognition probability exceeding 70%, indicating relatively low uncertainty.

We updated this results in Supplementary note line 158 to 160.

69 Line 308. "Samples" are difficult to discern when the figure is printed on paper.

Response: We enlarged page and uploaded the all-high-resolution figures.

70 Line 322. What is the reason for adding regression lines here? These lines indicated that the slope and the intercept increase with time. Could it be a symptom that the maps suffer from the quality of the Landsat data? In principle, forest patches of a certain size should always be mapped at a certain size +/- some uncertainty whereas here we observe a clear temporal trend.

Response: Here, we compared the planted forest area mapped in our study with the statistical area at provincial scale referred to (Wang et al., 2020). To present the correlation between our mapped area and statistical area, we re-drawn this figure, as shown in Fig. R6 (Supplementary Fig 3).

71 Line 326. What is shown in these maps?

Response: Here we show the area difference between our generated planted forest maps and forest inventory map with each 0.1-degree grid. Details to see the response to Comment 67.

Reference

- Calderón-Loor, M., et al. High-resolution wall-to-wall land-cover mapping and land change assessment for Australia from 1985 to 2015, *Remote Sens Environ.* 252, 112148 (2021).
- Cao, S. Why large-scale afforestation efforts in China have failed to solve the desertification problem. *Environ. Sci. Technol.* 42, 1826–1831 (2008).
- Cao, S., et al. Excessive reliance on afforestation in China's arid and semi-arid regions: Lessons in ecological restoration, *Earth Sci Rev.* 104, 240-45 (2011).
- Chen, S., et al. Current and future carbon stocks of natural forests in China, *For. Ecol. Manage.* 511: 120137 (2022).
- Khunrattanasiri, W. Application of Remote Sensing Vegetation Indices for Forest Cover Assessments, in *Concepts and Applications of Remote Sensing in Forestry*, M.N. Suratman, Editor. 2022, Springer Nature Singapore: Singapore. p. 153-166.
- Lai, L., et al. Carbon emissions from land-use change and management in China between 1990 and 2010, *Science Advances*, 2, e1601063 (2016).
- Li, R., et al. One of report of China's Six Key Forestry Programs: visiting of Sand Control Programs for areas in the vicinity of Beijing and Tianjin. *Forestry & Humans* 9, 14–18 (2001) (in Chinese).
- Liu, J., et al. Spatiotemporal characteristics, patterns and causes of land use changes in China since the late 1980s, *Dili Xuebao/Acta Geogr. Sin.*, 69, 3–14 (2014).
- Liu, W., et al. Uncertainty and confidence in land cover classification using a hybrid classifier approach. *Photogramm Eng Remote Sensing* 70(8), 963-971 (2004).
- Loosvelt, L., et al. Random Forests as a tool for estimating uncertainty at pixel-level in SAR image classification. *Int J Appl Earth Obs Geoinf.* 19, 173-184 (2012).
- Olofsson, P., et al. Good practices for estimating area and assessing accuracy of land change. *Remote Sens Environ.* 148, 42–57 (2014).
- Peng, S., et al. Afforestation in China cools local land surface temperature, *Proc. Natl. Acad. Sci. U.S.A.* 111, 2915-19 (2014).
- Rizayeva, A., et al. Large-area, 1964 land cover classifications of Corona spy satellite imagery for the Caucasus Mountains. *Remote Sens Environ.* 113343 (2023).
- State Forestry and Grassland Administration of China. Development report for the Three-North Shelterbelt System in the past 40 Years: 1978–2018 (2019).
- Wang, G., et al. China's forestry reforms. *Science* 318, 1556–1557 (2007).

- Wang, J., et al. Mapping sugarcane plantation dynamics in Guangxi, China, by time series Sentinel-1, Sentinel-2 and Landsat images, *Remote Sensing of Environment*, 247, 111951 (2020).
- Xu, B., et al. Biomass carbon stocks in China's forests between 2000 and 2050: A prediction based on forest biomass-age relationships, *Sci China Life Sci.* 53, 776-83 (2010).
- Yang, J. & Huang, X. The 30m annual land cover dataset and its dynamics in China from 1990 to 2019. *Earth Syst. Sci. Data* 13, 3907-3925 (2021).
- Yang, Q., et al. Mapping high-resolution forest aboveground biomass of China using multisource remote sensing data, *GIsci Remote Sens.* 60, 2203303 (2023).
- Yang, Z., et al. Loss of soil organic carbon following natural forest conversion to Chinese fir plantation, *For. Ecol. Manage.* 449, 117476 (2019).
- Zhai, D., et al. Lost in transition: Forest transition and natural forest loss in tropical China', *Plant Diversity* 39, 149-53 (2017).
- Zhang, X. et al. GLC_FCS30: global land-cover product with fine classification system at 30 m using time-series Landsat imagery. *Earth Syst. Sci. Data* 13, 2753-2776 (2021).
- Zhou, J., et al. Reconstruction of global MODIS NDVI time series: Performance of Harmonic ANalysis of Time Series (HANTS). *Remote Sens Environ.* 163, 217-228 (2015).

Reviewer #2:

This could be a data paper, but it falls short of a research paper targeting carbon stock assessment. The author used Landsat TM images and forest samples to generate planted forest maps in China from 1990 to 2020. Despite the fact that I believe the planted forest maps do add new knowledge to our understanding, the focus of this study - carbon stock assessment - was too simplified to be convincing.

Response: Thank you for your valuable feedback. Your insightful comments have been instrumental in enhancing the quality of our work. We have carefully considered each of your comments and made substantial revisions to the manuscript. Specially, we have strengthened the overall theme of the article and added information about the methods and results of carbon stock estimation, along with quantifying and analyzing the errors. Below, we provide comprehensive point-by-point responses to address the specific issues raised during the review process.

1 First, the carbon density of each vegetation type was obtained from the study of Lai et al (2016) (note that it is not Li et al, I believe). This inherits the uncertainties from the study (Lai's study used mean values for the same vegetation type compiled from different studies). Besides, forest types are not fixed over time. Instead, they evolve and change during succession. Using a fixed type to extrapolate the carbon potential will be biased.

Response: Thank you for your insightful and professional comments. Firstly, we sincerely apologize for the error in naming; it was indeed the study conducted by Lai et al published in 2016. In order to estimate the carbon storage of planted forests during various periods accurately, we utilized National Forest Inventory Data from 2004-2008 (representative of the year 2005) to derive carbon densities for different types of planted forests across different times using age-biomass density equations proposed by Xu et al (2010). Then, we estimated the planted forest carbon change in aboveground biomass by comparing storage in patch locations with the same vegetation type (Fig. 2), with the specific process as outlined below:

First-Acquisition of planted forest inventory samples for different forest types. Extracting planted forest samples from the National Forest Inventory Data based on the forest type annotations. Dividing the planted forest samples into 17 subsets (Supplementary Table 5) according to the forest types within vegetation map.

Second-Acquisition of planted forest samples across various periods. Obtaining plantation inventory samples for different vegetation types corresponding to different periods, based on age information from inventory samples and remote sensing time-series analysis, which entails two distinct scenarios: One scenario pertains to 1990-2000, where samples are directly filtered using the age of trees in inventory data; for instance, the 1990 samples are those inventory samples with a tree age exceeding 15 years, mirroring the forest age data from 2005 in our records. The second concerns samples from 2010-2020, utilizing time-series change analysis via remote sensing to assess the changes in each 2005 sample through to 2010, 2015, and 2020, preserving those unchanged and discarding any that have altered. By employing these methodologies, we acquire inventory samples for each period, inclusive of their tree species and age details.

Third-Estimation of carbon density. With the inventory sample collections established for each period, the age-biomass density equation (Eq 1) at the tree species scale crafted by (Xu et al. 2010) is applied to compute the biomass density for each sample under all forest types, and the carbon density value for each sample is determined using the 0.5 biomass-carbon conversion factor suggested for China's forests by (Ma et al., 2002). Subsequently, the average carbon density and its standard deviation for each type is calculated. It is important to note that this approach might overstate the carbon density of newly established planted forests between 2005-2020, as their ages range from 1-15 years, and the sample ages for 2010, 2015, and 2020 determined by the aforementioned method are all above 5 years. Consequently, for the newly added planted forests in this period, the study adopts the widely used space-for-time substitution approach (Blois et al. 2013) (Blois et al. 2013), selecting 2005 samples aged 1-15 years to estimate the carbon density of these new varied types of planted forests using the age-biomass equation (Eq 1).

$$B = \frac{w}{1 + ke^{-at}} \quad 1$$

In which B is the biomass density (Mg C/ha), t is the tree age (a), w, k, and a are constants that were determined by (Xu et al. 2010).

Fourth-Calculation of carbon storage in planted forests. According to forest types of vegetation map, we first classify planted forest into 17 forest types based on the principle of nearest neighbor in geography. Utilizing the carbon densities of these

vegetation types corresponding to disparate periods as previously established (Supplementary Table 5), the carbon storage of Chinese planted forests for these periods is calculated as:

$$C = \sum_{i=1}^N D_i \times Area_i$$

in which C is the total carbon storage of planted forest in biomass, D_i and $Area_i$ are the carbon density and area of the i^{st} planted forest type. N is the number of types.

Fifth-Validation. In order to demonstrate the reasonableness of the calculated carbon storage in planted forests, we employed publicly accessible field survey samples created by (Xu et al., 2019) (download from National ecological data center resource sharing service platform of <http://www.nesdc.org.cn/>) and China's forest biomass data generated by (Yang et al., 2023) (download from the website of <https://www.3decology.org/2023/08/02/china-forest-agb-map2019/>) for result validation. The comparison results indicated an R^2 ranging from 0.61 to 0.68 (Supplementary Fig. 14).

We supplemented the above method in line 486-541. Using these methods, we updated our carbon estimation results, and re-analysis their dynamics, please see line 142-248.

We have to admit that using the same vegetation type can bias the estimates. Therefore, we compared the classification results with updated 2020 China vegetation map (Su et al., 2020), which revealed that over 70% grids had the same vegetation types with the old vegetation type map generated by 1980s to 2000s (Supplementary Fig. 17). Moreover, we compared carbon storage estimation results using two different vegetation maps and found differences of only about 2% grids greater than 0.1 Tg C (Supplementary Fig. 17). Although the comparative results indicate minimal differences, it is inevitable that they will have some impact on the outcomes. In our revised manuscript, we have conducted a thorough discussion and analysis regarding the reasons for the errors. Please refer to line 542-614 for more details.

2 Second, using tree height as a proxy to project carbon density is also problematic. Indeed, there is a close relationship between tree height and P-PET. However, tree height is determined by site index, and factors such as soil fertility, slope, and aspect

are important. Moreover, height is also related to the planting density of the trees. Using a simplified relationship of height and P-PET to further extrapolate the carbon stock potential is highly unreliable.

Response: We agree. In consolidating feedback from various reviewers, we recognize that this section indeed bears limited relevance to our study and fails to emphasize the article's main topic. We also agree with the reviewers' remarks on this section. As a result, we have abbreviated this portion, placing increased emphasis in the manuscript on the changes in carbon storage due to the expansion of planted forests in China. Particularly by adding descriptions of carbon storage estimates and their dynamic characteristics over various periods and regions, and the carbon storage alterations induced by the conversion of diverse land cover types to planted forests. Details to see the response to Comment 1.

3 Because of the above-mentioned problems, I highly doubt the reliability of the discussion and conclusions. For example, in lines 322-325, this is not reasonable as there are no such absolute differences between planted and natural forests. An instance is that human interventions to expedite the succession process via complementary planting of seedlings, selective cutting, and pruning in natural secondary forests will also help increase carbon stock. Are those forests natural or planted?

Response: Following the reviewers' suggestions, we have supplemented the methods and results of carbon estimation, quantified potential errors, and made revisions and improvements to the entire manuscript, please refer to the responses to Comment 1 and 2. Additionally, we have included definitions of planted and natural forests in this study, narrowing the focus of our research; please see lines 371-379.

Reference

- Lai, L., et al. Carbon emissions from land-use change and management in China between 1990 and 2010, *Science Advances*, 2, e1601063 (2016).
- Xu, B., et al. Biomass carbon stocks in China's forests between 2000 and 2050: A prediction based on forest biomass-age relationships, *Sci China Life Sci.* 53, 776-83 (2010).
- Ma Q., et al. Carbon content rate in constructive species of main forest types in

northern China. *Journal of Beijing Forestry University* 24(5/6), 96-100 (2002) (in Chinese).

Blois, J.L., et al., Space can substitute for time in predicting climate-change effects on biodiversity. *Proc. Natl. Acad. Sci. U.S.A.* 110(23), 9374-9379 (2013).

Xu, L., et al. A dataset of carbon density in Chinese terrestrial ecosystems (2010s). *China Scientific Data*, 4(1), p. 7 (2019).

Yang, Q., et al., Mapping high-resolution forest aboveground biomass of China using multisource remote sensing data. *Gisci Remote Sens.* 60(1), p. 2203303 (2023).

Su, Y., et al. An updated Vegetation Map of China (1:1000000). *Sci. Bull.* 65(13): p. 1125-1136 (2020).

Reviewer #3:

Comment for authors:

This manuscript obtained planted forest maps for each five-year interval from 1990 to 2020 based on Landsat data and random forest algorithm, and analyzed the sources of planted forest expansion in China. Meanwhile, this study quantified the C storage resulting from the expansion of planted forest over the past 30 years, and evaluate the biomass C storage potential of the additional planted forest. Overall, this study is truly unprecedented, carrying milestone significance in enhancing our understanding of the temporal and spatial changes within China's planted forests and their pivotal role in mitigating global climate change. However, there are some minor issues that need to be solved before publication.

Response: We greatly appreciate the positive evaluation of our work by the reviewer. In accordance with the reviewer's suggestions, we have made revisions and improvements to the entire manuscript.

Main Comments:

1、 The manuscript mentions that canopy height is a significant factor related to forest carbon storage potential. However, it appears that the author did not take canopy height into account when calculating the carbon storage conversion from other land types to planted forests. Instead, the author calculated carbon storage solely based on the conversion area. This oversight raises questions about the accuracy of the carbon storage estimate.

Response: Thank you for your professional suggestions. First, by integrating the comments of the reviewers, we have reduced the content on canopy height predicting carbon storage potential, and enhanced the analysis of carbon storage changes caused by the expansion of planted forests in China (Line 142-236 of the Results section). Secondly, to accurately measure the changes in carbon storage, we calculated the carbon density for each vegetation type in each period based on forest age, biomass density equations proposed by Xu et al (2010), National Forest Inventory Data, and China's 1:1,000,000 vegetation map (Zhang, 2007). By incorporating age to quantify changes in carbon density, and ultimately, using Lai et al.'s approach, estimating the above-ground carbon storage and its variations in planted forests over various periods (Line 450-505 of the Method section). Simultaneously, we conducted an in-depth

analysis of the errors present in carbon storage estimation, providing a more objective conclusion for the results of this study (Line 542-614 of the Method section).

2、 In light of the points mentioned above, we assert that the results pertaining to P-PET in the final section (Lines 202-219) have limited relevance to the overall content. Their removal will not disrupt the paper's logical structure but will enhance its conciseness.

Response: We agree and we delete this section and strengthened the results of carbon storage in planted forest biomass. Please see the revised manuscript in Line 142-248.

3、 In the analysis of Land Use and Land Cover (LULC) transfer, the author identifies shrubland, grassland, and natural forest as the primary sources of plantation forest. While the author provides comprehensive explanations and references for shrubland and grassland in the discussion, there appears to be a gap in addressing the key issue of natural forest loss, which is arguably the most critical forest type to protect. It is evident that a more in-depth discussion regarding the reasons behind the loss of natural forests is warranted.

Response: Following your suggestions, we concluded several reasons about the loss of natural forest. The first aspect is policy-driven. Beginning in the 1970s and 1980s, China extensively harvested natural forests to meet national and economic development demands, leading to a severe decline in natural forest quality over decades (Cao et al. 2011). In efforts to rejuvenate natural forest ecosystems, the state enhanced their structure via afforestation, resulting in the conversion of a substantial part of low-quality natural forest vegetation into artificial forests (Cao et al. 2011). Secondly, to fulfill economic requirements like forest products, especially in China's subtropical areas, numerous natural forests have been transformed into fir plantations (Yang et al. 2019). Research indicates that in numerous southern regions in China, the cultivation of fast-growing and high-yielding economic planted forests has resulted in substantial natural forest depletion (Zhai et al. 2017). Given revised manuscript focusing on carbon storage changes resulting from planted forest expansion, we have opted not to elaborate excessively on land conversion, instead using land conversion as a means to elucidate carbon storage variations. Taking into account the integration of the reviewers' kind comments and suggestions, this study concentrates on the carbon storage changes associated with the expansion of planted forests. To better emphasize the theme, the study does not provide an overly detailed explanation of

land conversion, but rather uses land conversion to explain the changes in planted forest carbon storage.

4、 The quality and consistency of the two vegetation-type maps are crucial factors influencing the reliability of the results concerning carbon storage changes. I have concerns regarding the consistency in classifying vegetation types within these two maps. If different classification systems were employed, it would render them non-comparable. Therefore, it is imperative that the author provides more detailed information about these two vegetation maps.

Response: The classification systems of these two vegetation maps are the same.

Moreover, the new version was updated based on the old one using a massive number of ground survey samples; thus, they share the same classification system (Su et al., 2020). We also compared the differences in vegetation types corresponding to planted forests between the two vegetation maps, which revealed that over 70% grids had the same vegetation types between the old vegetation type map generated by 1980s to 2000s and new updated vegetation map in 2020 (Supplementary Fig. 17). Moreover, we compared carbon storage estimation results using two different vegetation maps and found differences of only about 2% grids greater than 0.1 Tg C (Supplementary Fig. 17). We added these descriptions in the revised manuscript line 584-614. Consequently, we utilized the older version of the vegetation map, combined with carbon density estimates for different periods, to calculate the carbon storage of planted forests in various periods, and analyzed their spatiotemporal variation characteristics and reasons.

Minor Comments:

4、 Line 290, native forest should be natural forest;

Response: Thanks, we deleted this section and focused on the discussion of carbon storage change in planted forest biomass. Also, in order to avoid grammatical errors, we have used a professional polishing agency to polish the full text, and provided a proof of polishing.

5、 Line 304 underscores should be underscore.

Response: Thanks, and we corrected it. We have also checked and polished the full text, and provided proof of polishing

6、 The contents of subgraph b, and d of Fig 3 are not corresponding with the title of the subgraph.

Response: Thanks, by incorporating the reviewer's feedback, we have removed Figure 3, and added information about carbon analysis. Please see Fig.3-5.

7、 Line 541, it is suggested to add respectively at the end of the sentence to increase readability.

Response: Thanks, and we added it, please see Line in 446

8、 Line 502, the full name of the FMASK algorithm should be given at the first time.

Response: The full name of FMASK was supplemented in Line 390-391.

9、 Line 539 to 541, the classification model adapted to each region and period should be provided.

Response: We utilized a random forest model for classification, employing consistent parameters for the model. The difference lies in the distinct classification features inputted. The input feature set for different periods and regions are provided in the Supplementary; please see Supplementary Table 5. We have also referenced Supplementary Table 5 in the main text; please see line 444.

10、 Line 557, work should be works.

Response: Thanks, and we corrected it, please see Line in 542

11、 Line 587, delete word code.

Response: Thanks, and we corrected it, please see Line in 470

12、 Line 589, vegetation-type maps should be vegetation-type map.

Response: Thanks, and we corrected it, please see Line in 487.

Reference

Cao, S., et al. Excessive reliance on afforestation in China's arid and semi-arid regions: Lessons in ecological restoration, *Earth Sci Rev.* 104, 240-45 (2011).

Su, Y., et al. An updated Vegetation Map of China (1:1000000). *Sci. Bull.* 65(13): p. 1125-1136 (2020).

- Xu, B., et al. Biomass carbon stocks in China's forests between 2000 and 2050: A prediction based on forest biomass-age relationships, *Sci China Life Sci.* 53, 776-83 (2010).
- Yang, Z., et al. Loss of soil organic carbon following natural forest conversion to Chinese fir plantation, *For. Ecol. Manage.* 449, 117476 (2019).
- Zhai, D., et al. Lost in transition: Forest transition and natural forest loss in tropical China', *Plant Diversity* 39, 149-53 (2017).
- Zhang, X. S. *Vegetation Map of the People's Republic of China (1:1,000,000) and Its Illustration Put to Press*, *Acta Ecologica Sinica* (2007).

REVIEWER COMMENTS

Reviewer #1 (Remarks to the Author):

Dear Authors

The manuscript has been revised thoroughly and according to the comments, for which I am very thankful. All explanations make sense, and the manuscript has become much clearer. Accordingly, the results are in a proper context and well understood. The manuscript will therefore make a valid contribution to Nature Communications. Nonetheless, I still have some remarks that I share with the authors in this review. I would appreciate if these could be clarified in a new revision. In particular, the way errors are characterized is not sufficiently addressed yet and validation of the C dynamics is questionable.

Once such basic aspects are clarified, the authors may consider whether to address an additional request, which may further enhance the value of this study. Would be of benefit to add information of C dynamics for the same time period for natural forests? I let the authors consider this option.

Comments

Line 32. The 637.2 TgC are added to what?

Line 44. A reference to the Agenda 2030 is appreciated. Does the Agenda 2030 specifically address planted forests or forests in general?

Line 45. Since the topic is planted forest, I am wondering whether the "land" spatiotemporal dynamics rather than the "forest" spatiotemporal dynamics should be referred to.

Line 119. Could you please clarify in brackets what is the size of the pixel?

Line 120. Please rephrase "positive trends in area increase of planted forests".

Lines 123-127. My apologies but the sentences on these lines are not clear to me.

Line 128. "The decline trend in the rate of planted forest area expansion" contains too many directions (trend, rate, expansion). Please rephrase.

Line 135. Referring to Fig. 1 a, b and c, is the pixel size 10 x 10 km?

Line 146. What is the meaning of 16.2 TgC?

Line 154. Fig. 3 shows an increase but how can one distinguish between C accumulated by existing planted forest and newly planted forests?

Line 163. I believe that it is the "C storage" of planted forests.

Line 178. "Change" means "difference", right?

Line 185. What is the definition of "standard deviation"? Is it precision?

Line 253. I would replace "of planted forests C storage" with "terrestrial C storage through planted forests".

Lines 300-302. Could you at least add a reference and possibly some numbers from this "previous research"?

Line 305. All forests? I understood that Figures 2 and 3 are about planted forests only so the meaning of "total" below is not clear.

Line 372. I might have missed this, but what is the definition of forest in this study? My understanding is that when seedlings are planted, they do not count as forest yet but that may just be a matter of definition.

Line 468. Does "pixel" stand for grid cell?

Lines 487-511. I would appreciate a revision of the text and streamline. In particular lines 490-494 are difficult to comprehend.

Line 526. Does “tree” here stand for “forest”?

Lines 535-541. Looking at the two datasets cited here, I have doubts about the overall validation. The title of the dataset by Xu et al., suggests that the data was collected around the year 2010. The map-based estimates of AGB by Yang et al., are representative of 2019-2021. None of them appear to have a temporal component which makes the validation of the magnitude of the C trajectories questionable. In addition, the Wang et al. dataset appears to be overall underestimating AGB (Fig. 7 therein), which makes even the assessment in Fig. S14 questionable. In addition, Fig. S14 reports TgC which would mean that the comparison is done at an aggregated scale. I would very much appreciate some clarifications here. As such, these assessments do not qualify as validation of the results of this study.

Lines 542-614. The addition of a section on uncertainties is very much appreciated. It is obvious that all details cannot be clarified but the text now is only scratching the surface of a possibly delicate issue: how large are the uncertainties associate with this study. I am missing a quantitative evaluation of the errors (bias and precision), in particular in the validation of the C estimates and the C trends.

Line 577-578. Why is the uncertainty based on the max probability of classification? Would this not underplay the real uncertainty?

Line 582. A histogram instead of multiple illustrations very similar to each other would be more illustrative.

Line 594. Are planted trees of the same species of other trees already growing in the same area?

Line 597. Where does one see that 70% of the grids have the same vegetation type in Fig.

S17?

Line 599. The caption of Fig. S19 is unclear to understand the text here.

Reviewer #2 (Remarks to the Author):

I feel that the manuscript has been greatly improved. However, I think there are still a few points that need to be clarified. The most important one is to clarify the definition of forest. Could you give a more specific definition of the forest? For example, what was the canopy coverage threshold adopted in developing the forest maps? It was also reported that China also included a specific type of shrubland as forest, as well as economic plantations and bamboo. This is essential because it determines the reliability of your comparisons between the forest area derived from the maps and the statistics reported in the national survey.

Other suggestions:

1. In lines 119-125, suggest adding resample process here. These results were derived from a map resampled from 30*30 m resolution, right? This should be clarified because you previously mentioned that the maps were from Landsat at 30 m resolution (boolean type data), but here you talked about trends. I was confused until I read the content in Line 464. Please provide details here.
2. Line 515, this is minor but there are species-specific conversion factors. Please check and see if there are differences if species-specific conversion factors used.
3. In the calculation of the biomass density, I am confused. Did the author use the equations developed by Xu et al or the equations were developed using survey data? Since the authors have substantial samples, it is strange to use the equations developed by Xu et al.
4. The reliability of the entire manuscript was built on the forest maps developed previously. Did the results of the carbon stock change (e.g. Lines 30-32 in Abstract) include the uncertainties from forest mapping? Did the authors conduct grid-to-grid comparisons at 0.1 degree or at 30m resolution?

Reviewer #3 (Remarks to the Author):

Thank you for your diligent consideration of the review comments. Your thorough revisions not only showcase a profound understanding of the research but also reflect a commitment to academic excellence. Your proactive response to the review comments and the extensive modifications made have significantly enhanced the quality of the manuscript, providing readers with more comprehensive and lucid information.

I genuinely admire the effort you have invested in the revision process, believing that such dedication will undoubtedly garner greater recognition and impact for your research within the academic community.

Once again, thank you for your hard work and positive response to the review comments.

Response to Reviewers

Reviewer #1

The manuscript has been revised thoroughly and according to the comments, for which I am very thankful. All explanations make sense, and the manuscript has become much clearer. Accordingly, the results are in a proper context and well understood. The manuscript will therefore make a valid contribution to Nature Communications. Nonetheless, I still have some remarks that I share with the authors in this review. I would appreciate if these could be clarified in a new revision. In particular, the way errors are characterized is not sufficiently addressed yet and validation of the C dynamics is questionable.

Once such basic aspects are clarified, the authors may consider whether to address an additional request, which may further enhance the value of this study. Would be of benefit to add information of C dynamics for the same time period for natural forests? I let the authors consider this option.

Response: We are grateful for your thorough review and constructive feedback on our manuscript. We are pleased to hear that the revisions have enhanced the clarity and context of our study and that the results are now well understood. We appreciate your acknowledgment of the potential contribution of our work to Nature Communications.

Regarding your remarks, we agree that further clarification is necessary regarding the characterization of errors and the validation of carbon dynamics. Firstly, to quantify uncertainty, we referred to the method used by Chazdon et al. (2016) and calculated the mean and standard deviation of the carbon density of each forest type based on the forest inventory samples. The mean value was used to estimate carbon storage, and then we calculated the total uncertainty by calculating the carbon storage in planted forests using the lower and upper standard deviation values to provide a measure of uncertainty associated with the total carbon storage of planted forests. We added a detailed explanation about the measurement of uncertainty in Lines 615-620.

Secondly, to further validate the carbon estimation results, this study once again compared the estimated carbon storage for 2020 with the aboveground biomass data released by the European Space Agency (Please see Lines 535-544).

About the suggestion on C dynamics of natural forests, we firstly thank you for your valuable suggestions, and we agree with your perspective. However, the main focus of this study is to address the carbon dynamics of plantation forests. Therefore, to maintain thematic consistency and focus, we did not address issues related to natural forests. In future research, we will prioritize your suggestion to assess the carbon

dynamics of natural forests. Thank you again for your professional and insightful suggestions.

Comments

1 Line 32. The 637.2 TgC are added to what?

Response: This means that the expansion of planted forest area from 1990 to 2020 resulted in an increase of 637.2 TgC in the carbon storage of aboveground biomass. We added an explanation in Lines 32-33.

2 Line 44. A reference to the Agenda 2030 is appreciated. Does the Agenda 2030 specifically address planted forests or forests in general?

Response: Thanks for the suggestion, we added a reference to the Agenda 2030. Please see Line 44. The Agenda 2030 addressed forests.

3 Line 45. Since the topic is planted forest, I am wondering whether the “land” spatiotemporal dynamics rather than the “forest” spatiotemporal dynamics should be referred to.

Response: We agree with your comment, and replaced “forest” using “land”. Please see Line 46.

4 Line 119. Could you please clarify in brackets what is the size of the pixel?

Response: We supplemented the explanation about the size of the pixel, please see Lines 119-121.

5 Line 120. Please rephrase “positive trends in area increase of planted forests”.

Response: We simplified the sentence while keeping the original meaning unchanged “We found that 91% of the changed pixels from 1990 to 2020 showed positive trends”. Please see Line 121-122.

6 Lines 123-127. My apologies but the sentences on these lines are not clear to me.

Response: Here, we analyzed the trend of the rate of change in planted forest areas. We rephrased this sentence, please see Lines 124-128.

7 Line 128. “The decline trend in the rate of planted forest area expansion” contains too many directions (trend, rate, expansion). Please rephrase.

Response: We rephrased this sentence, please see Lines 124-128.

8 Line 135. Referring to Fig. 1 a, b, and c, is the pixel size 10 x 10 km?

Response: The size of the pixel is 0.1°. We added this explanation in the caption of Figure 1.

9 Line 146. What is the meaning of 16.2 TgC?

Response: This represents the standard deviation of carbon estimation. To avoid misunderstanding we added an explanation in Line 146.

10 Line 154. Fig. 3 shows an increase but how can one distinguish between C accumulated by existing planted forest and newly planted forests?

Response: Thank you for your comment. We constructed two carbon density databases (Supplementary Table 6) based on the relationship between forest age and biomass density in the forest inventory data, using a spatial substitution for time method (Please see details in Methods Lines 488-5424). This includes the carbon density of newly planted forests and stable forests at different time periods. Based on this carbon density database, we estimated the carbon storage brought by newly planted forests.

11 Line 163. I believe that it is the “C storage” of planted forests.

Response: Agree, we revised this sentence. Please see Lines 163-164

12 Line 178. “Change” means “difference”, right?

Response: We unified it to changes. Please see Line 178

13 Line 185. What is the definition of “standard deviation”? Is it precision?

Response: The standard deviation here is derived from multiple carbon density calculations corresponding to each forest type. To quantify uncertainty, we referred to the method used by Chazdon et al. (2016) and calculated the mean and standard deviation of the carbon density of each forest type based on the forest inventory samples. The mean value was used to estimate carbon storage, and then we calculated the total uncertainty by calculating the carbon storage in planted forests using the lower and upper standard deviation values to provide a measure of uncertainty associated with the total carbon storage of planted forests. We added a detailed explanation about the measurement of uncertainty in Line 615-620.

14 Line 253. I would replace “of planted forests C storage” with “terrestrial C storage through planted forests”.

Response: Thanks for the suggestion. We agreed and replaced using “terrestrial C storage through planted forests”. Please see Lines 253-254

15 Lines 300-302. Could you at least add a reference and possibly some numbers from this “previous research”?

Response: We added references. Please see Line 302.

16 Line 305. All forests? I understood that Figures 2 and 3 are about planted forests only so the meaning of “total” below in not clear.

Response: We corrected this sentence. It is “China’s planted forests”. Please see Line 306.

17 Line 372. I might have missed this, but what is the definition of forest in this study?

My understanding is that when seedlings are planted, they do not count as forest yet but that may just be a matter of definition.

Response: Thank you for your valuable suggestion. In this study, forests are defined as areas with tree height exceeding 5 meters and canopy cover exceeding 15% within a 30-meter resolution pixel. This definition is inherited from CLCD land cover data, as the plantation forests in this study are extracted within the forest cover range of the CLCD time series. We provided detailed explanations about the forest definition in the manuscript. Please see Lines 373-374.

18 Line 468. Does “pixel” stand for grid cell?

Response: Yes. To clarify more clearly, we replaced “pixel” with “grid”. Please see Line 470.

19 Lines 487-511. I would appreciate a revision of the text and streamline. In particular lines 490-494 are difficult to comprehend.

Response: Thanks for the comment. We re-write this section and tried our best to make it more clearly. Please check in lines 489-496.

20 Line 526. Does “tree” here stand for “forest”?

Response: Yes, we revised using “forest”. Please Line 524.

21 Lines 535-541. Looking at the two datasets cited here, I have doubts about the overall validation. The title of the dataset by Xu et al., suggests that the data was collected around the year 2010. The map-based estimates of AGB by Yang et al., are representative of 2019-2021. None of them appear to have a temporal component which makes the validation of the magnitude of the C trajectories questionable. In addition, the Wang et al. dataset appears to be overall underestimating AGB (Fig. 7 therein), which makes even the assessment in Fig. S14 questionable. In addition, Fig. S14 reports TgC which would mean that the comparison is done at an aggregated scale. I would very much appreciate some clarifications here. As such, these assessments do not qualify as validation of the results of this study.

Response: Thank you for your professional comments. We agree with your point that it is indeed difficult to obtain field survey data that truly represents a specific time, and there is a lack of time-series samples for validation. Therefore, in the validation phase, this study only validated the results estimated for 2010 and 2020. Firstly, we verified the carbon storage results of plantation forests in 2010 using field survey data published around 2010 by Xu et al. and verified the results for 2020 using aboveground biomass data produced by Yang et al. for 2019. Additionally, to make the validation robust, we again utilized the 2020 global aboveground biomass data product released by the European Space Agency. The validation results showed significant consistency between our study results and the European Space Agency's data product, with an R^2 of 0.5. Overall, the accuracy of carbon storage estimation in this study ranged from 0.5 to 0.7. We provided further supplementation and explanations on the validation process in the manuscript Line 535-544.

In addition, we added an illustration that the comparison is done at an aggregated scale in the caption of Fig S14.

22 Lines 542-614. The addition of a section on uncertainties is very much appreciated. It is obvious that all details cannot be clarified but the text now is only scratching the surface of a possibly delicate issue: how large are the uncertainties associated with this study. I am missing a quantitative evaluation of the errors (bias and precision), in particular in the validation of the C estimates and the C trends.

Response: Thanks for the comment. Please see the response to **Comment 13**

23 Line 577-578. Why is the uncertainty based on the max probability of classification? Would this not underplay the real uncertainty?

Response: Thanks for your professional comment. Random forests, as an ensemble learning method, improve model accuracy and robustness by constructing multiple decision trees and aggregating their predictions (Breiman, 2001). The final classification result is typically based on the "majority vote" principle, where the classification supported by the most decision trees is chosen (Breiman, 2001). Therefore, the maximum probability, which is the proportion of decision trees that select a certain classification as the final result, is often used to quantify uncertainty in the classification of remote sensing (Pal, 2005). For example, Loosvelt et al. (2012) used this method to assess the uncertainty of random forest classification. Therefore, we selected this method to quantify the uncertainty of classification.

However, the method is not without its limitations, which could lead to underestimating the true uncertainty in some scenarios. For example, in cases of class imbalance, where some classes dominate the dataset, the maximum probability might bias towards these classes, even for samples where the model's true certainty is lower (Cutler et al., 2007). Additionally, samples lying close to decision boundaries between classes might exhibit high maximum probabilities, masking the inherent uncertainty due to their proximity to multiple class regions.

In the future, supplementary measures can be employed to provide a more nuanced view of uncertainty. For example, considering the entropy across all class probabilities can offer insights into the overall uncertainty level, with higher entropy indicating greater uncertainty. Similarly, examining the distribution of votes among classes, rather than focusing solely on the majority class, can reveal cases where the model is almost equally split between two or more classes, indicating higher uncertainty (Rokach, 2016).

24 Line 582. A histogram instead of multiple illustrations very similar to each other would be more illustrative.

Response: Thanks for the suggestion. We added a histogram figure in Supplementary Figure 16 as shown below:

25 Line 594. Are planted trees of the same species of other trees already growing in the same area?

Response: It's similar, and here we refer to the study by Yu et al. (2020).

26 Line 597. Where does one see that 70% of the grids have the same vegetation type in Fig. S17?

Response: Sorry, this is a clerical error, it should be over 80%. We calculated the carbon storage in each grid and compared their differences to judge whether the vegetation type within the grid has changed. Because the area of plantation forests is the same in each grid if the vegetation types are different, their carbon densities will also differ, leading to differences in carbon storage between them; if there is no difference, it indicates that the vegetation type has not changed. We counted the number of carbon storage differences equal to 0, which was approximately 80.3%, and annotated it in the legend.

27 Line 599. The caption of Fig. S19 is unclear to understand the text here.

Response: The reference here is to Supplementary Figure 17, and we have modified the expression in the original text. Here, we want to illustrate that only 2.1% of the estimated differences in carbon storage brought by the two vegetation maps exceed 0.1 Tg. Please see Lines 598-601.

Reference

1. Loosvelt, L., Peters, J., Skriver, H., Lievens, H., Van Coillie, F. M. B., Baets, B. D., and Verhoest, N. E. C., 2012. Random Forests as a tool for estimating uncertainty at pixel-level in SAR image classification, *International Journal of Applied Earth Observation and Geoinformation*. 19, 173-184.
2. Cutler, D. R., Edwards, T. C., Beard, K. H., Cutler, A., Hess, K. T., Gibson, J., & Lawler, J. J. (2007). Random forests for classification in ecology. *Ecology*, 88(11), 2783-2792.
3. Rokach, L. (2016). Decision Forest: Twenty Years of Research. *Information Fusion*, 27, 111-125.
4. Breiman, L. (2001). Random Forests. *Machine Learning*, 45(1), 5-32.
5. Pal, M. (2005). Random forest classifier for remote sensing classification. *International Journal of Remote Sensing*, 26(1), 217-222.
6. Transforming our world: the 2030 Agenda for Sustainable Development (United Nations, 2015); <https://go.nature.com/3CsPxqn>
7. Chazdon, R. L., Broadbent, E. N., Rozendaal, D. M. A., Bongers, F., Zambrano, A.

M. A., Aide, T. M., Balvanera, P., Becknell, J. M., Boukili, V., Brancalion, P. H. S., Craven, D., Almeida-Cortez, J. S., Cabral, G. A. L., de Jong, B., Denslow, J. S., Dent, D. H., DeWalt, S. J., Dupuy, J. M., Durán, S. M. et al. (2016). Carbon sequestration potential of second-growth forest regeneration in the Latin American tropics. *Science Advances*, 2(5), e1501639.

Reviewer #2:

I feel that the manuscript has been greatly improved. However, I think there are still a few points that need to be clarified. The most important one is to clarify the definition of forest. Could you give a more specific definition of the forest? For example, what was the canopy coverage threshold adopted in developing the forest maps? It was also reported that China also included a specific type of shrubland as forest, as well as economic plantations and bamboo. This is essential because it determines the reliability of your comparisons between the forest area derived from the maps and the statistics reported in the national survey.

Response: Thank you for your valuable suggestion. In this study, forests are defined as areas with tree height exceeding 5 meters and canopy cover exceeding 15% within a 30-meter resolution pixel. This definition is inherited from CLCD land cover data, as the plantation forests in this study are extracted within the forest cover range of the CLCD time series. We provided detailed explanations about the forest definition in the manuscript. Please see Lines 373-374.

Other suggestions:

1. In lines 119-125, suggest adding resample process here. These results were derived from a map resampled from 30*30 m resolution, right? This should be clarified because you previously mentioned that the maps were from Landsat at 30 m resolution (boolean type data), but here you talked about trends. I was confused until I read the content in Line 464. Please provide details here.

Response: Thanks a lot for the suggestion, we added the description of the resampling process. Please check in lines 119 -121.

2. Line 515, this is minor but there are species-specific conversion factors. Please check and see if there are differences if species-specific conversion factors used.

Response: We thank you and agree with your comment. This study mainly referred to the recommendations provided by Fang et al (2021). in their book “Carbon Budgets of Forest Ecosystems in China”, and used a conversion coefficient of 0.5 more suitable for Chinese forest ecosystems. According to Chen (2003), the average carbon content of tree species in China is greater than 0.45. Using 0.45 as the conversion coefficient would result in an underestimation of the biomass carbon pool. Therefore, using 0.5 as the conversion coefficient for forest biomass carbon may be more reasonable.

3. In the calculation of the biomass density, I am confused. Did the author use the equations developed by Xu et al or the equations were developed using survey data? Since the authors have substantial samples, it is strange to use the equations developed by Xu et al.

Response: Thanks for your valuable comment. We directly used the models established by Xu et al. because the data we used to establish carbon density were the same as theirs.

4. The reliability of the entire manuscript was built on the forest maps developed

previously. Did the results of the carbon stock change (e.g. Lines 30-32 in Abstract) include the uncertainties from forest mapping? Did the authors conduct grid-to-grid comparisons at 0.1 degree or at 30m resolution?

Response: Thanks for your insightful comment. We agree with your point. This study validated the accuracy of the plantation forest map through various commonly used methods, with an overall accuracy ranging from 77.3% to 81.8% and good consistency with statistical data, with an R^2 between 0.8 and 0.9, indicating the reliability of the data products obtained in this study. Therefore, based on such results, we conducted carbon storage estimation research. In the process of carbon storage estimation, we did not quantify the uncertainty of the planted forest maps but mainly considered the influence of carbon density. However, it is undeniable that the accuracy of plantation forest mapping does affect the estimation of carbon storage. Based on your valuable suggestion, we will consider further quantifying the uncertainty of plantation forests in future research.

The comparisons and analyses in this study were performed within 0.1-degree grid cells.

Reference

1. Chen X. 2003. Study on the carbon sink function of major forest types in North China. Beijing Forestry University (in Chinese).
2. Fang J., Zhu J., Li P. 2021. Carbon Budgets of Forest Ecosystems in China. Science Press (in Chinese).

Reviewer #3:

Thank you for your diligent consideration of the review comments. Your thorough revisions not only showcase a profound understanding of the research but also reflect a commitment to academic excellence. Your proactive response to the review comments and the extensive modifications made have significantly enhanced the quality of the manuscript, providing readers with more comprehensive and lucid information.

I genuinely admire the effort you have invested in the revision process, believing that such dedication will undoubtedly garner greater recognition and impact for your research within the academic community.

Once again, thank you for your hard work and positive response to the review comments.

Response: Thank you for your positive feedback on our revised manuscript. I look forward to my research receiving more attention from our peers and contributing valuable insights to our field. Thank you again!

REVIEWERS' COMMENTS

Reviewer #1 (Remarks to the Author):

The manuscript underwent a second through revision and all my questions have been addressed. Looking forward to seeing the article published.